



# Accounting for Black Carbon Aging Process in a Two-way Coupled Meteorology - Air Quality Model

Yuzhi Jin[1,2], Jiandong Wang[1, 2, *], David C. Wong[3,4], Chao Liu [1,2], Golam Sarwar[3], Kathleen M. Fahey[3], Shang Wu[1,2], Jiaping Wang[5,6], Jing Cai[7], Zeyuan Tian[1,2], Zhouyang Zhang[1,2], Jia Xing[8], Aijun Ding[5,6], and Shuxiao Wang[9]

[1] China Meteorological Administration Aerosol-Cloud and Precipitation Key Laboratory, School of Atmospheric Physics, 1 Nanjing University of Information Science and Technology, Nanjing, 210044, China
[2] Collaborative Innovation Center on Forecast and Evaluation of Meteorological Disasters, Nanjing University of Information Science and Technology, Nanjing, 210044, China
[3] US Environmental Protection Agency, Research Triangle Park, NC, 27711, USA
[4] Department of Earth and Atmospheric Sciences, University of Houston, Houston, 77204, USA
[5] Joint International Research Laboratory of Atmospheric and Earth System Sciences, School of Atmospheric Sciences, Nanjing University, Nanjing, 210023, China
[6] National Observation and Research Station for Atmospheric Processes and Environmental Change in Yangtze River Delta, Nanjing, 210023, China
[7] Institute for Atmospheric and Earth System Research, Faculty of Science, University of Helsinki, Helsinki, 00014, Finland
[8] Department of Civil and Environmental Engineering, University of Tennessee, Knoxville, 37996, USA
[9] State Key Joint Laboratory of Environment Simulation and Pollution Control, School of Environment, Tsinghua University, Beijing, 100084, China

*Correspondence to:* Jiandong Wang (jiandong.wang@nuist.edu.cn)

**Abstract.** Black carbon (BC) exerts significant impacts on both climate and environment. BC aging process alters its hygroscopicity and light absorption properties. Current models, like the Weather Research and Forecasting - Community Multiscale Air Quality (WRF-CMAQ) two-way coupled model, inadequately characterize these alterations. In this study, we accounted for BC aging process in the WRF-CMAQ model (WRF-CMAQ-BCG). We introduced two new species (Bare BC and Coated BC) in the model and implemented a module to simulate the conversion from Bare BC to Coated BC, thereby characterizing the aging process. Furthermore, we improved the cloud chemistry and aerosol optics modules to analyze the effects of BC aging on hydrophobicity and light absorption. The simulated results indicate a spatial distribution pattern with Bare BC prevalent near emission sources and Coated BC more common farther from sources. The average Number Fraction of Coated BC ($NF_{coated}$) is approximately 57%. Temporal variation exhibits a distinct diurnal pattern, with $NF_{coated}$ increasing during the daytime. The spatial distribution of wet deposition varies significantly between Bare and Coated BC. Bare BC exhibits a point-like deposition pattern, whereas Coated BC displays a zonal distribution. Notably, Coated BC dominates the BC wet deposition process. Additionally, incorporating BC aging process reduces BC wet deposition by 17.7% and increases BC column concentration by 10.5%. The simulated Mass Absorption Cross-section ($MAC$) value improved agreement with observed measurements. Overall, the WRF-CMAQ-BCG model enhances the capability to analyze aging-related variables and BC mixing state, while also improving performance in wet deposition and optical properties.



## 1 Introduction

Black carbon (BC) refers to the aerosol emitted into the atmosphere from incomplete combustion of carbonaceous fuels. As a predominant light-absorbing aerosol, BC exhibits strong absorption capability within the visible wavelength range, surpassing other light-absorbing aerosols (Bond et al., 2013). This pronounced light-absorbing property has significant impacts on climate and air quality (Tan et al., 2020). It influences direct radiative forcing by absorbing and scattering sunlight. Also, it produces indirect radiative forcing by influencing cloud formation or altering surface albedo when deposited on ice or snow surfaces (Hansen and Nazarenko, 2004; Jacobson, 2004; Ramanathan and Carmichael, 2008). Additionally, BC aerosol can exacerbate regional air pollution by changing local meteorological conditions. The light-absorbing property of BC leads to atmospheric heating, depressing the development of the planetary boundary layer (PBL) and promoting the formation of regional atmospheric pollution (Ding et al., 2013; 2016). Moreover, the unique morphology of BC aerosol, which provides a large surface area, facilitates heterogeneous reactions with trace gases, thereby promoting the formation of secondary particulate matter (Zhang et al., 2020). Therefore, BC aerosol is of great significance in both meteorological and environmental fields.

The BC component exhibits very low chemical reactivity and is refractory (Bond et al., 2013). Consequently, previous studies have typically considered BC aerosol as chemically inert, primarily serving as a reaction interface for other chemical reactions due to its unique morphology (Monge et al., 2010). However, freshly emitted BC aerosol experiences condensation, coagulation, and heterogeneous oxidation processes during atmospheric transport, becoming coated by scattering aerosol components and converted into aged BC aerosol (Riemer at al., 2009; Oshima and Koike, 2013; Tan et al., 2020). During this transformation, a significant part of BC aerosol changes from an external to an internal mixing state (Pratt et al., 2011; Stevens and Dastoor, 2019). BC aging process alters the hydrophobicity and light-absorbing property of BC aerosol, thereby affecting cloud condensation nuclei (CCN) concentration, wet deposition, and aerosol optical properties (Yu, 2000; Oshima and Koike, 2013; Bond et al., 2006). Therefore, when evaluating the impact of BC aerosol in meteorology and environmental fields, BC aging process cannot be neglected.

Numerical simulations of BC aerosol have gradually incorporated the effects of BC aging process. Various aerosol modeling approaches, each with distinct characteristics, have evolved along different developmental trajectories. These models are mainly divided into four categories: bulk model, modal model, sectional model, and particle-resolved model. The bulk model primarily categorizes BC aerosol into hydrophobic and hydrophilic types, conceptualizing the aging process as a conversion from hydrophobic to hydrophilic BC. The global 3-D chemical transport model (GEOS-Chem) is a good representative of the bulk model approach. The default GEOS-Chem model uses a fixed e-folding lifetime for this conversion (Park et al., 2003). Huang et al. (2013) added a condensation-coagulation scheme, which is affected by water vapor, ozone, hydroxyl radical and sulfuric acid concentration, in the GEOS-Chem model. He et al. (2016) developed a "hybrid" microphysics-based aging scheme for condensation, coagulation, and heterogeneous chemical oxidation processes in the GEOS-Chem model. While the details are continuously refined, these schemes fundamentally represent BC aging process as a conversion from hydrophobic to hydrophilic BC. The modal model mainly addresses the aging process by representing its different aging states of BC within



different modes. An example of this paradigm is the Community Atmosphere Model version 5 - Modal Aerosol Module

(CAM5-MAM). The three-mode version (MAM3) places BC aerosol in the accumulation mode, assuming it is fully internally mixed. To account for BC aging process, both the four-mode version (MAM4) (Liu et al., 2016) and the seven-mode version (MAM7) (Liu et al., 2012) use BC in the primary carbon mode to represent freshly emitted BC, which converts to aged BC in the accumulation mode after the aging process. The sectional model expands traditional size bins to two-dimensional size-BC proportion bins for considering BC mixing state during the aging process. For instance, the default Weather Research and

Forecasting model coupled with Chemistry (WRF-Chem) includes BC aerosol concentration, without BC mixing state. Matsui et al. (2013) developed a size and BC mixing state resolved model based on the WRF-Chem model (MS-resolved WRF-Chem model) that divides BC within each size bin based on mass fraction to describe changes in BC mixing state during the aging process. A representative example of the particle-resolved model is the Particle Monte Carlo (PartMC) model. When coupled with the Model for Simulating Aerosol Interactions and Chemistry (PartMC-MOSAIC), it can precisely characterize the

chemical composition of each particle, thus accurately tracking the aging process of BC aerosol. These four models have integrated respective techniques to deal with BC aging process and each model execution time has increased accordingly. Each technique has its own benefits and drawbacks as well as additional computational burden. Therefore, selecting appropriate methods to account for BC aging process and striking a balance between accuracy and computational burden remains an ongoing exploration.

The Community Multiscale Air Quality (CMAQ) model, developed by the US Environmental Protection Agency (EPA), is widely used in the research community as well as in the US government as a regulatory model and is continuously evolving. In the CMAQ model, the mixing state of BC aerosol is simplified to a fully internal mixing state. This assumption has limited impact on the offline air quality model which focuses on species' mass concentration, because the mass of BC is unchanged before and after the aging process. However, when considering the impact of aerosols on meteorology, this is likely to change.

The properties of aerosols, including their aging process, play a critical role in how they influence radiative effects and CCN, necessitating a more comprehensive approach in modeling their effects. The Weather Research and Forecasting - Community Multiscale Air Quality (WRF-CMAQ) two-way coupled model is designed to account for the interactive two-way feedback between aerosols and meteorological conditions (Wong et al., 2013). Considering BC aging process is necessary due to its substantial impact on changes in hydrophobicity and light absorption, which significantly affect the WRF-CMAQ coupled

model. In this study, we accounted for BC aging process in the WRF-CMAQ model (WRF-CMAQ-BCG) by introducing two new species and constructing a BC aging module. Furthermore, we improved the cloud chemistry module and aerosol optical module to investigate the effects of changes in hydrophobicity and light absorption caused by BC aging process. Section 2 describes the overall construction of the WRF-CMAQ-BCG model and the development methods for each module. Section 3 provides the case description used for simulation and the model evaluation. The performance of the new model regarding

aging-related variables, BC mixing state, BC wet deposition, and aerosol optics is presented in Sect. 4.





## 2 The WRF-CMAQ-BCG Model

### 2.1 New Model

During the aging process, BC aerosol becomes coated with scattering aerosol components, converting from an externally mixed state to an internally mixed state. This process converts hydrophobic BC into hydrophilic BC, enabling it to act as cloud condensation nuclei (CCN), thereby influencing the aerosol indirect radiative effect. Concurrently, the coating acquired during aging enhances BC's absorption properties through the lensing effect (Lack and Cappa, 2010). This amplification of absorption directly impacts overall radiation dynamics, underscoring the profound influence of BC aging on both direct and indirect aerosol-radiation interactions in the atmosphere (the WRF-CMAQ model does not yet have the indirect radiative effect capability). The original WRF-CMAQ model does not account for BC aging process (Fig.1, left panel). The model assumes that freshly emitted BC aerosol undergoes instantaneous aging, resulting in a uniformly fully internally mixed state without spatiotemporal variation. To address this limitation, we developed the WRF-CMAQ-BCG model, capable of handling the main changes in BC aerosol properties due to aging throughout the entire atmospheric lifecycle, from emission to deposition (Fig.1, right panel).

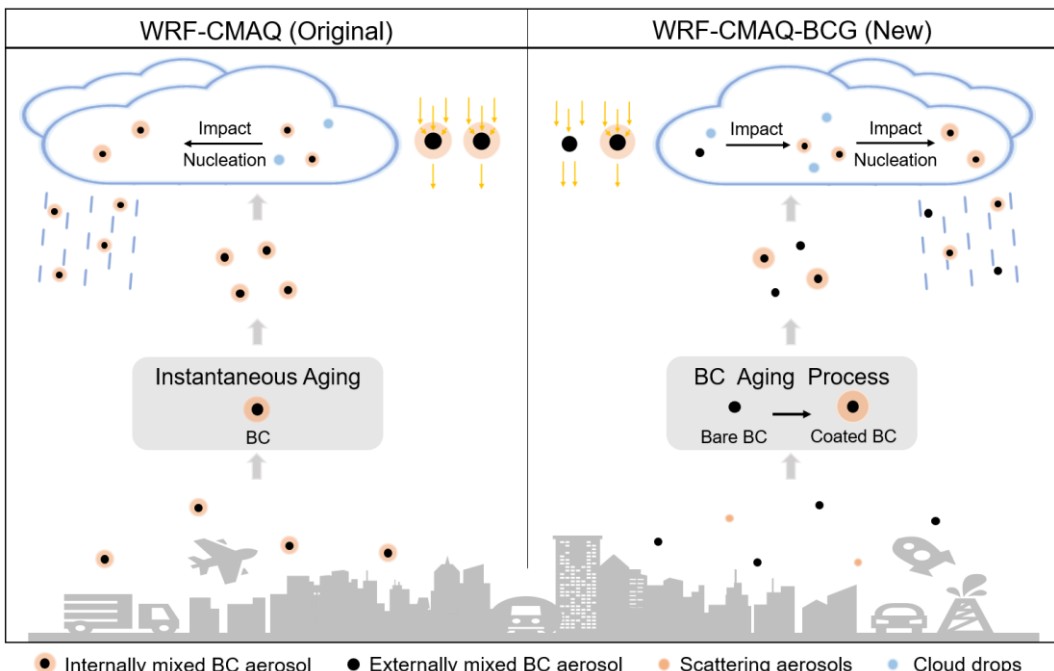

**Figure 1: The BC mixing state in the WRF-CMAQ model and the WRF-CMAQ-BCG model.**

To differentiate the aging states of BC aerosol, we introduced two new species: "Bare BC (ABEC)" and "Coated BC (ACEC)" in our model (Table 1). They replace the single "BC (AEC)" in the original model. Bare BC represents unaged, hydrophobic,





externally mixed BC aerosol, while Coated BC represents aged, hydrophilic, internally mixed BC aerosol. It is important to note that Bare BC and Coated BC are used to differentiate the aging states of BC aerosol, and their combination is equivalent to the BC in the original WRF-CMAQ model.

**Table 1. New species added in the model.**

|  | Species name | Mixing state | Hygroscopicity |
| --- | --- | --- | --- |
| Bare BC | ABEC | External | Hydrophobic |
| Coated BC | ACEC | Internal | Hydrophilic |


The WRF-CMAQ model provides meteorological and chemical conditions for BC aging process, while BC aging process in turn, influences the meteorology and air quality simulated by the model. This continuous and cyclic interaction in the WRF-CMAQ coupled model provides a realistic representation of the atmospheric processes. The primary differences between Bare BC and Coated BC lie in their hydrophobicity and light absorption property. Changes in hydrophobicity mainly affect aqueous

chemistry, scavenging and wet deposition, while changes in the light absorption property affect aerosol optical calculations. For other atmospheric processes, including advection, diffusion, turbulence and dry deposition, the behavioral differences between Bare BC and Coated BC are minimal. Consequently, the emission module, cloud chemistry module and aerosol optics module of the WRF-CMAQ-BCG model require relevant updates based on the inclusion of the BC aging process, as shown in Fig. 2. Freshly emitted BC aerosol (Bare BC) enters the BC aging module from the emission module and gradually converts

into Coated BC through the aging process. When Bare BC and Coated BC enter the cloud chemistry module, hydrophobic Bare BC cannot become CCN, so it cannot undergo nucleation scavenging during in-cloud scavenging, in turns affecting the wet deposition of BC aerosol. Bare BC exists in an external mixing state, whereas Coated BC has stronger light absorption due to the lensing effect. Therefore, Bare BC and Coated BC need to be considered separately when calculating aerosol optics. In addition, under the WRF-CMAQ two-way coupled model framework, changes in BC concentration stemming from

alteration in wet deposition impact aerosol optics calculations. Similarly, updates in the aerosol optics properties alter the radiative budget and also influence BC aging process.



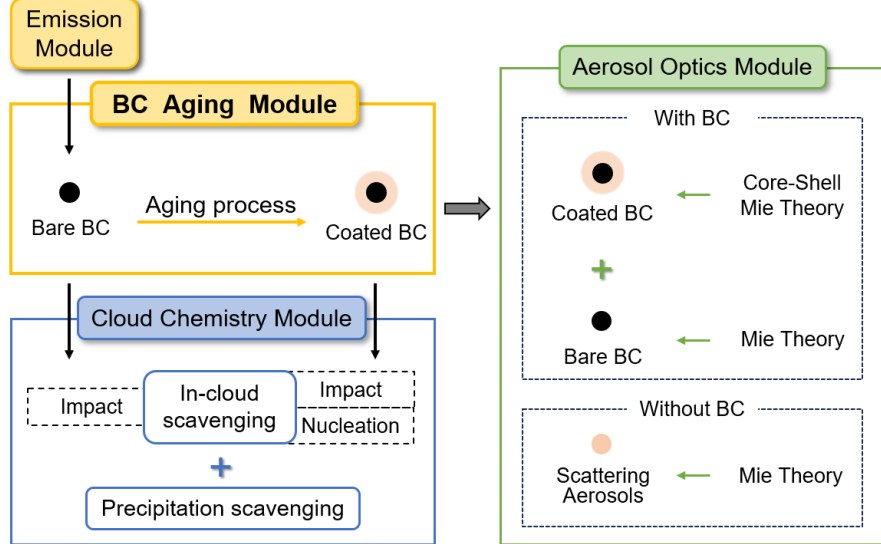

**Figure 2: Schematic diagram of the WRF-CMAQ-BCG model structure.**


## 2.2 BC Aging Module

In the WRF-CMAQ-BCG model, we constructed a specialized BC aging module to account for BC aging process in the atmosphere. The aging process is represented by the conversion from Bare BC to Coated BC. Development of appropriate aging schemes is crucial for accurately simulating this dynamic conversion process. The aging rate and the aging timescale are

two important variables related to the aging process. The aging rate indicates the speed of the aging process as shown in Eq. (1). While the aging timescale reflects the time required for the aging process, inversely related to the aging rate, as shown in Eq. (2). Global climate and atmospheric chemical transport models often employ a simplified approach to carbonaceous aerosol aging, typically using a uniform aging timescale of approximately 1 day (Huang et al., 2013). For two specific examples, the traditional GEOS-Chem model (Park et al., 2003) and the Regional Climate Model (RegCM) (Solmon et al., 2006) both set a

fixed conversion lifetime of 1.15 days. This constant aging scheme is computationally efficient but cannot capture the dynamic changes of the aging process. Consequently, some researchers have developed more sophisticated approaches, parameterizing the aging rate and linking it to specific atmospheric components (Huang et al., 2013; Croft et al., 2005). The hydroxyl (OH) radical, a key player in atmospheric chemistry, facilitates the conversion of sulfur dioxide ($SO_2$) to sulfuric acid ($H_2SO_4$). The condensation of $H_2SO_4$ gas onto the surface of particles leads to the conversion of carbonaceous aerosols from hydrophobic to

hydrophilic. Recognizing this relationship, some studies have used the OH radical to parameterize the aging rate of BC (Liu et al., 2011; Huang et al., 2013; Oshima and Koike, 2013).

The aging rate ($k$) and the aging timescale ($\tau$) are shown in Eq. (1) and Eq. (2):





$$k = \frac{\partial(M_{\text{coated}}) / \partial t}{M_{\text{bare}}}, \tag{1}$$

$$\tau = \frac{1}{k}, \tag{2}$$

where $M_{\text{bare}}$ and $M_{\text{coated}}$ are the mass concentration of Bare BC and Coated BC aerosol, respectively.

This study employs a BC aging module that quantifies the BC aging rate using an equation dependent on the concentration of OH radicals, as shown in Eq. (3). Condensation is considered through the setting of a fast-aging term, while coagulation is considered through a slow-aging term. Based on the selected OH aging scheme, the dynamic process of BC aging is represented by setting a virtual reaction, wherein Bare BC progressively converts into Coated BC.

$k = \beta \, [\text{OH}] + \alpha \,, \tag{3}$

where $k$ represents the aging rate, [OH] represents OH radical concentration. $\beta$ and $\alpha$ are assumed to be constant, with values $4.6 \times 10^{-12}$ cm$^3$ molecule$^{-1}$ s$^{-1}$ and $5.8 \times 10^{-7}$ s$^{-1}$, $\beta$ is estimated by assuming an e-folding aging timescale of 2.5 days, and $\alpha$ is estimated by assuming a 20 days e-folding lifetime for coagulation (Liu et al., 2011; Huang et al., 2013; Oshima and Koike, 2013).

## 2.3 Cloud Chemistry Module

The removal pathways for BC aerosol in the atmosphere are dry and wet deposition (Begam et al., 2016; Bibi et al., 2017), with wet deposition accounting for nearly 80% (Textor et al., 2006; Choi et al., 2020). While dry deposition is primarily governed by surface characteristics and meteorological conditions, showing minimal sensitivity to BC aging, this study prioritizes wet deposition processes. In the WRF-CMAQ model, the cloud chemistry module calculates in-cloud scavenging

and precipitation scavenging of BC aerosol, performs aqueous chemistry calculations, and accumulates wet deposition. Precipitation scavenging associated with precipitation processes such as rain, snow, and graupel, and is minimally affected by changes in BC aerosol properties before and after aging. In contrast, aerosol in-cloud scavenging which includes nucleation scavenging and impact scavenging, is strongly associated with BC aging process. Nucleation scavenging refers to the process where aerosols serve as CCN, becoming cloud droplets enveloped by cloud water and subsequently removed. Impact

scavenging occurs through collisions with cloud droplets (Barrett et al., 2019). The hydrophobicity changes caused by BC aging process mainly affect nucleation scavenging in the way that hydrophobic Bare BC cannot act as CCN, while hydrophilic Coated BC can. In other words, Bare BC cannot undergo nucleation scavenging. The WRF-CMAQ model further differentiates scavenging mechanisms based on particle size: BC aerosol in the accumulation mode undergoes nucleation scavenging, while BC aerosol in the Aitken mode experiences impact scavenging as interstitial aerosol (Binkowski and Roselle, 2003). When

Aitken mode Bare BC undergoes impact scavenging, it converts to Coated BC due to water envelopment, perpetuating its removal from the atmosphere, as illustrated in right panel of Fig. 1. Overall, these updates in the cloud chemistry module in this work enhance the representation of BC aerosol in various aspects.





## 2.4 Aerosol Optics Module

BC aerosol, a dominant light-absorbing atmospheric constituent, plays a crucial role in direct radiative forcing. As BC ages,
the BC component becomes coated by other components, which act like lenses, refracting more light onto the BC aerosol and
significantly enhancing its light absorption (Lack and Cappa, 2010). The WRF-CMAQ model assumes a fully internally mixed
state for BC, treating all BC aerosol as coated. This approach neglects the presence of externally mixed Bare BC in the
atmosphere, leading to an overestimation of the overall light absorption by aerosols. The third part of this work will address
this limitation.

In the WRF-CMAQ model, the light absorption of aerosols is entirely attributed to BC aerosol. BC aerosol is considered the
core, with water-soluble aerosols, insoluble aerosols, aerosol water, and sea salt as the shell. Each substance has its
corresponding refractive index across 14 wavelengths under the Rapid Radiative Transfer Model (RRTM) for global climate
model (GCM) applications (RRTMG) scheme. For aerosols containing BC in the aitken and accumulation modes, the Core-
shell Mie theory is employed to calculate their optical characteristics. In contrast, coarse mode aerosols, which lack BC content,
are simulated using the standard Mie theory. In our WRF-CMAQ-BCG model, we calculated the optics of Bare BC and Coated
BC separately. We introduced a variable, the number fraction of Coated BC ($NF_{coated}$). The $NF_{coated}$ variable was brought into
the aerosol optics module for translating BC core back to Bare BC core and Coated BC core. The shell only surrounds the
Coated BC core. The Coated BC is calculated using the Core-shell Mie theory, while Bare BC and purely scattering aerosols
are calculated using the Mie theory. By apportioning the BC core, the overestimation of aerosol light absorption can be
corrected.

$$NF_{coated} = \frac{N_{coated}}{N_{bare} + N_{coated}} , \qquad (4)$$

where $NF_{coated}$ represents the number fraction of Coated BC, $N_{bare}$ and $N_{coated}$ are the number concentration of Bare BC and
Coated BC aerosol, respectively. Bare BC and Coated BC are merely in different aging states, they are essentially BC aerosol
with the same density and volume. Therefore, the number fraction of Coated BC can be calculated from the mass fraction of
Coated BC.

## 3 Case description and model evaluation

To evaluate the performance of the new model and analyze BC aging process, we conducted a simulation using the WRF-
CMAQ-BCG model with WRF version 4.4.1 and CMAQ version 5.4 over the continental United States (CONUS) domain for
the entire month of June 2010. The model uses a horizontal spatial resolution of 12 km with 35 vertical layers. A one-month
spin-up period (May 2010) was utilized to initialize the model. The initial and boundary conditions were configured with equal
proportions of Bare BC and Coated BC. The WRF model parameterization schemes selected for the simulation case are listed
in Table 2. The CMAQ model employs the "cb6r5_AERO7" chemical mechanism and utilizes the Rosenbrock solver.





Additionally, the model assimilated data from three sources: Final Operational Global Analysis (FNL), North American
Mesoscale Forecast System (NAM), and North American Regional Reanalysis (NARR) datasets, provided by the National
Weather Service's National Centers for Environmental Prediction (NCEP)/National Center for Atmospheric Research (NCAR).

**Table 2. Setup of physical process schemes in WRF model simulation.**

| Physics processes | Schemes | References |
|---|---|---|
| Micro physics | Morrison 2–moment Scheme | Morrison et al., 2009 |
| Planetary Boundary Layer Physics | Asymmetric Convection Model 2 Scheme | Pleim, 2007 |
| Cumulus Parameterization | Kain–Fritsch Cumulus Potential Scheme | Kain, 2004 |
| Shortwave and Longwave | RRTMG Shortwave and Longwave Schemes | Iacono et al., 2008 |
| Land Surface | Pleim–Xiu Land Surface Model | Pleim and Gilliam, 2009 |
| Surface Layer | Pleim–Xiu Scheme | Pleim, 2006 |

Figure 3 illustrates the spatial distribution of BC emission on the simulation domain in red and the blue star represents the
ground observation station T0 of the Carbonaceous Aerosols and Radiative Effects Study (CARES) campaign, located in
Sacramento, California. The observation dataset is available from the Atmospheric Radiation Measurement (ARM) program
of the US Department of Energy (DOE) (https://adc.arm.gov/discovery/#/results/s::CARES). This comprehensive campaign
collected a diverse array of data, encompassing aerosols, atmospheric conditions, cloud properties, and radiation data. The
accuracy of the dataset has been widely recognized (Zaveri et al., 2012; Cahill et al., 2012). Data collected from this site is
used as the benchmark for evaluation and comparison. The data used in our study are presented in Table 3.





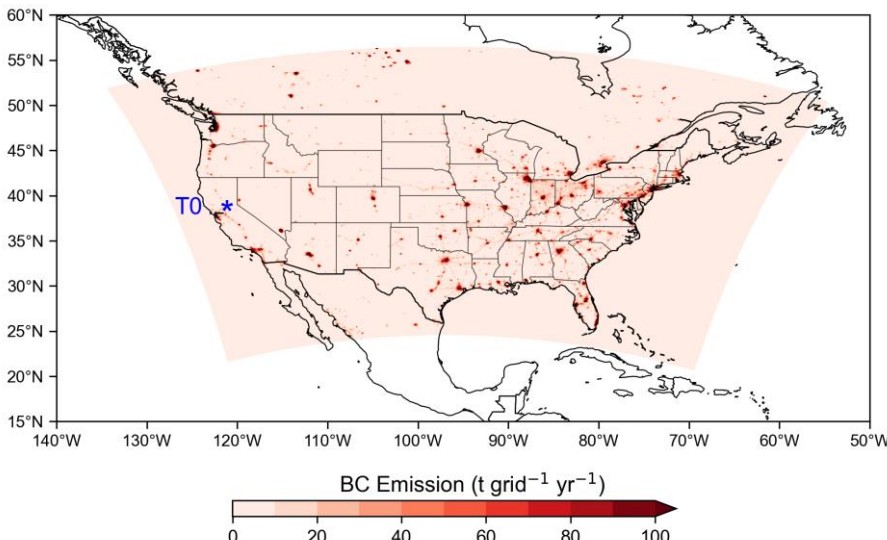

**Figure 3: Spatial distribution of BC emission for June 2010 on the model simulation area (Red Color). The blue star indicates an observation site, T0. (38.6483° N, 121.3493° W).**


**Table 3. Description of observation data.**

| Data type | Data | Data description | Instrument/Technique |
|---|---|---|---|
| Meteorology | T | Temperature | Vaisala WXT-510 |
| | RH | Relative humidity | Vaisala WXT-511 |
| | P | Pressure | Vaisala WXT-512 |
| | WS | Wind speed | Vaisala WXT-513 |
| | WD | Wind direction | Vaisala WXT-514 |
| Gas | $O_3$ | $O_3$ concentration | UV absorption |
| | $SO_2$ | $SO_2$ concentration | Thermo Model 43i |
| | NO | NO concentration | Chemiluminescence |
| | $NO_2$ | $NO_2$ concentration | Photolytic conversion CU GMAX-DOAS |
| BC | $M_{BC}$ | BC mass concentration | SP2 |
| Optics properties | $b_{abs}$ | Absorption coefficient | PAS |

To evaluate the accuracy of the model, we compared the simulation results of the WRF-CMAQ model and the WRF-CMAQ-BCG model with various meteorological observations, as well as the volumetric concentrations of several gases at the T0 site





in the CARES campaign. Figure 4 illustrates the comparison of meteorological data, including temperature, relative humidity, atmospheric pressure, wind direction, and wind speed. The resulting Mean Bias Error (*MBE*) for these variables were 0.14 K, 1.14%, 0.79 hPa, 0.50 m/s, and 28.92° in the WRF-CMAQ model and 0.13 K, 1.19%, 0.79 hPa, 0.50 m/s, and 28.33° in the WRF-CMAQ-BCG model, respectively. Figure 5 presents the model and observation comparison of four gases: $O_3$, $SO_2$, NO, and $NO_2$ at the T0 site, based on the *MBE* metric. The resulted values for these gases compared to the observations were being

-7.83 ppbv, 0.077 ppbv, 1.42 ppbv, and 2.91 ppbv in the WRF-CMAQ model and -7.87 ppbv, 0.074 ppbv, 1.43 ppbv, and 2.91 ppbv in the WRF-CMAQ-BCG model, respectively. These results demonstrate that both the WRF-CMAQ model and the WRF-CMAQ-BCG model exhibit good accuracy and have insignificant differences. Clearly, incorporating BC aging process into the WRF-CMAQ coupled model does not compromise the overall model performance and it can provide reasonably accurate meteorological and chemical conditions for the aging process.


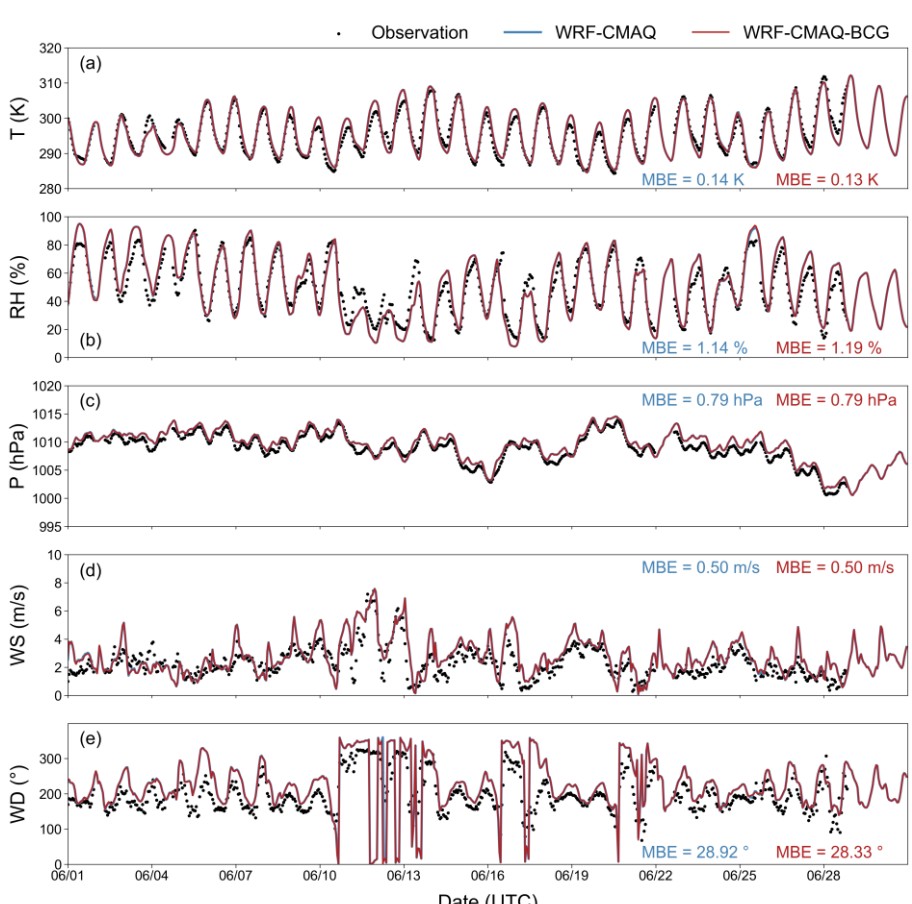

**Figure 4: Comparison of simulated and observed meteorological data. (a) Temperature, (b) Relative humidity, (c) Pressure, (d) Wind speed, and (e) Wind direction.**





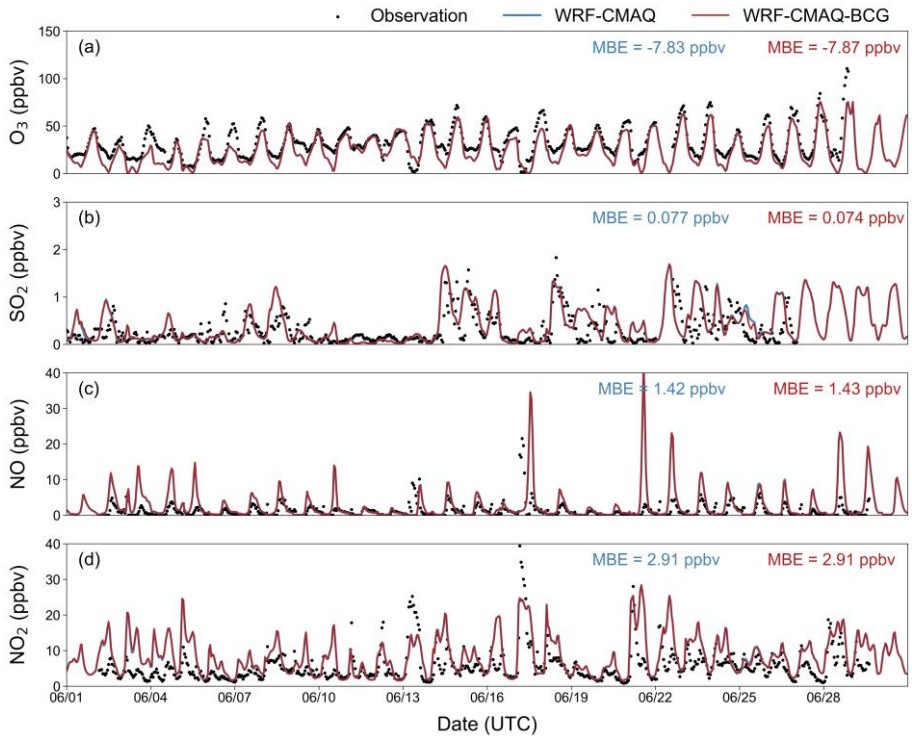


**Figure 5: Comparison of simulated and observed several gases concentration. (a) O₃ concentration, (b) SO₂ concentration, (c) NO concentration, and (d) NO₂ concentration.**

# 4 Results

## 4.1 Aging-related variables

The aging rate ($k$) and the aging timescale ($\tau$) are important variables to quantify the aging process. The standard WRF-CMAQ model lacks the capability to generate BC aging-related variables. In contrast, in the WRF-CMAQ-BCG model, spatiotemporal variations of the aging-associated variables in the BC aging process are related to the concentration of OH radicals. Figure 6(a) shows the aging rate, with higher values in the central region corresponding to the dark orange areas and lower values in the southeastern and western parts corresponding to the yellow areas. The average value is $2.26 \times 10^{-5}$ s$^{-1}$. The spatial distribution

of the aging timescale shown in Fig. 6(b) exhibits a reciprocal pattern, with an average aging timescale of 17.49 hours, which is close to the value of 18 hours obtained by Peng in Houston (Peng et al., 2016).



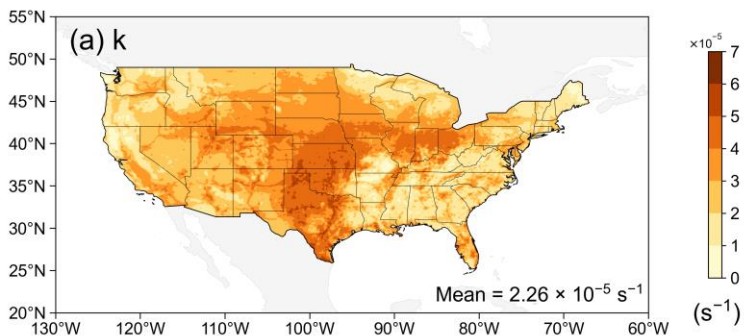

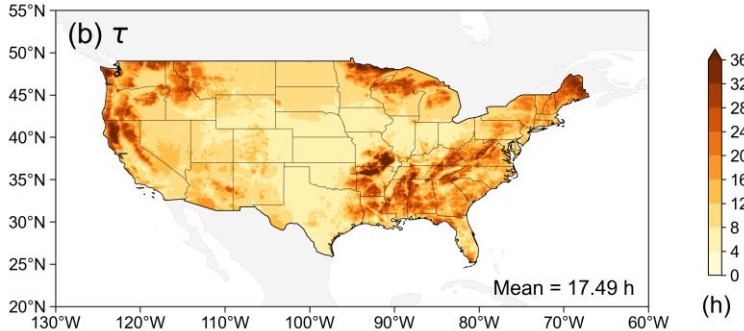

**Figure 6: Spatial distribution of aging-related variables (average hourly values for June). (a) the aging rate and (b) the aging**
**timescale.**

## 4.2 Mixing State

Mixing state is a key microphysical characteristic of BC aerosol and a major feature that undergoes significant changes in aging process. As aging progress, BC transforms from a completely externally mixed state to a more complex configuration where internally mixed and externally mixed BC coexist. Therefore, we can use the number fraction of Coated BC ($NF_{coated}$)
to represent the mixing state of BC aerosol.

Regarding the spatial distribution of BC aerosol, as illustrated in Fig. 7, the simulated results of the WRF-CMAQ-BCG model demonstrate that high mass concentrations of Bare BC and Coated BC are primarily distributed in the eastern United States and the West Coast region. This can be attributed to the higher population density and the presence of heavy industrial and commercial activities in those regions, resulting in a notable increase in BC emissions (Fig. 3). The simulated mean mass
concentration of Bare BC in the U.S. region is 0.045 μg m$^{-3}$, while the mean mass concentration of Coated BC is 0.034 μg m$^{-3}$. The simulated mean $NF_{coated}$ in the U.S. region is 57%. The spatial distribution of mixing state shows that Bare BC predominates in the eastern United States and the West Coast, while Coated BC is more prevalent in most of the western region. This phenomenon arises from the more extensive distribution of BC emissions in the eastern region and the West Coast, leading to a dominance of Bare BC. In contrast, in the western region (excluding the West Coast), where BC emissions are less
widespread, aging and transport processes play a more significant role, resulting in a higher proportion of Coated BC. In





summary, the BC mixing state exhibits a distinct pattern: Bare BC predominates in proximity to emission sources, while Coated BC becomes more prevalent in regions distant from these sources, a result from transport.

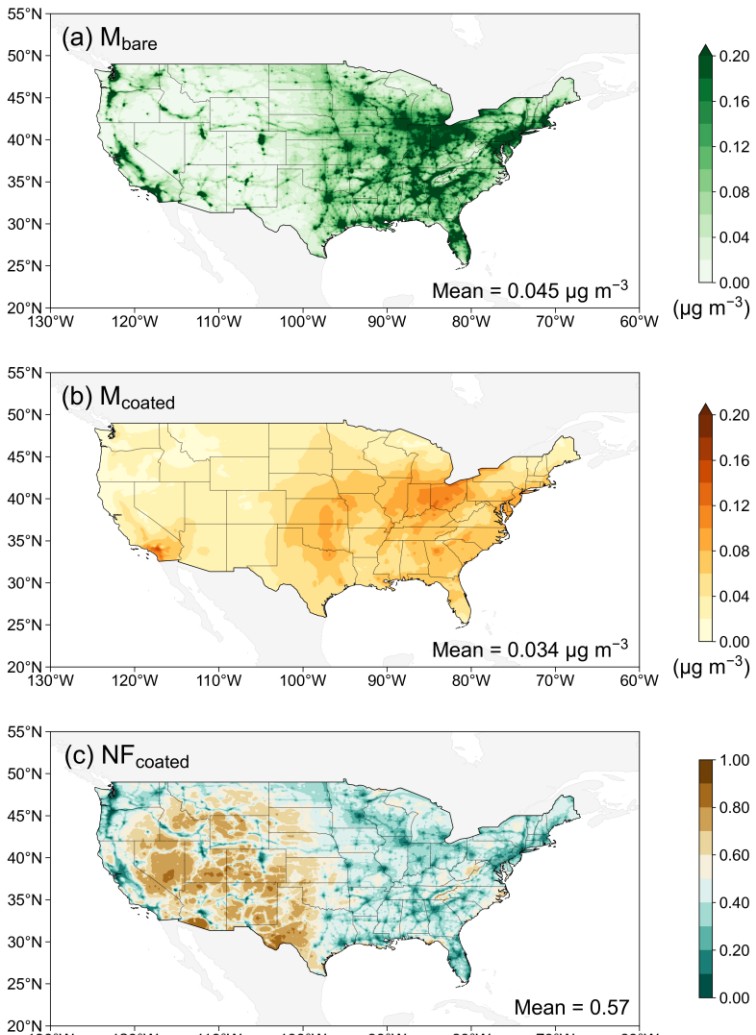

**Figure 7: Spatial distribution of (a)Bare BC mass concentration, (b) Coated BC mass concentration, and (c) the number fraction of Coated BC, with average hourly values for June.**

We examined the BC mixing state at the T0 site, for exploring the temporal variations of BC aerosol. Figure 8 presents the daily average variation of BC mixing states at the T0 site. It should be noted that this site is near the emission sources, illustrated
by elevated mass concentrations and a higher proportion of Bare BC. Figure 8 also shows that the mass concentration of Bare BC noticeably decreases in the afternoon, while the mass concentration of Coated BC significantly increases during this time. This pattern can be attributed to two primary factors. Firstly, the diurnal variation in traffic emissions, which typically peak





during morning and evening rush hours. Secondly, the intense solar radiation during the daytime enhances photochemical reactions (Peng et al., 2016), thereby accelerating BC aging and increasing Coated BC concentrations during the day. As a

result, the number fraction of Coated BC simulated shows a distinct increase from 8:00 to 20:00 local time, peaking in the afternoon. This finding is similar to the conclusions of Shen et al., 2023. Many research studies also have observed a noticeable increase in the fraction of Coated BC during the daytime, as shown in Fig. 8(d) (Lan et al., 2013; Huang et al., 2012; Wang et al., 2014, 2016, 2018; Zhang et al., 2018). The black line represents the mean of other observational studies, the gray area shows their range, revealing prominent diurnal variations with a higher daytime contribution. Therefore, the BC mixing state

reflects a temporal variation characteristic, with the proportion of Coated BC significantly increasing during the daytime period.

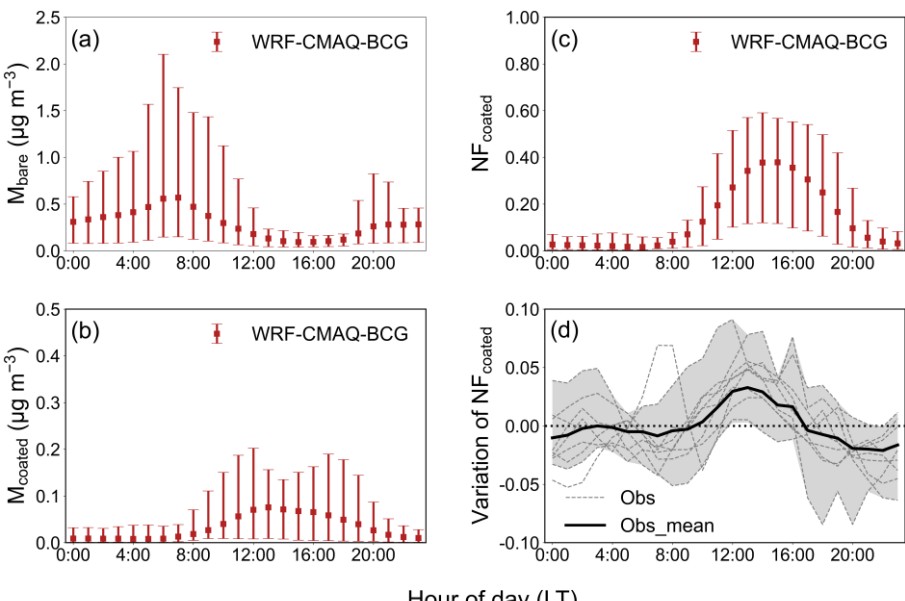

**Figure 8: Comparison of daily average variation in simulated results, (a) Bare BC mass concentration, (b) Coated BC mass concentration, (c) Number fraction of Coated BC ($NF_{coated}$), (d) Daily average variation of $NF_{coated}$ in other observational studies.**

**4.3 Wet Deposition**

Wet deposition is an important mechanism for the removal of BC aerosol from the atmosphere. The aging process significantly alters the hydrophobic nature of BC aerosol, consequently affecting their susceptibility to in-cloud scavenging. The quantity of BC wet deposition serves as a prominent indicator of this effect. The distribution of Bare BC wet deposition exhibits a characteristic point-like pattern in Fig. 9(a). This distinctive distribution is attributed to the short atmospheric lifetime of Bare

BC, which mainly undergoes aging to become Coated BC in less than three days. Consequently, Bare BC tends to be deposited before it can spread widely, resulting in a localized, point-like distribution. In contrast, Coated BC, which undergoes in-cloud scavenging and can be transported more widely by atmospheric processes, exhibits a zonal distribution. These contrasting deposition patterns provide valuable insights into the lifecycle and transport mechanisms of BC aerosol in the atmosphere. The



point-like distribution of Bare BC wet deposition underscores the rapid aging and displaying a localized impact of fresh BC emissions, while the zonal distribution of Coated BC wet deposition highlights the role of aging in enhancing BC's atmospheric mobility and its potential for long-range transport. Additionally, the average wet deposition of Bare BC is $1.31 \times 10^{-7}$ mg m$^{-2}$ d$^{-1}$, whereas the average wet deposition of Coated BC is $3.02 \times 10^{-2}$ mg m$^{-2}$ d$^{-1}$, differing by five orders of magnitude. Therefore, we can conclude that the wet deposition of BC aerosol is primarily due to Coated BC.

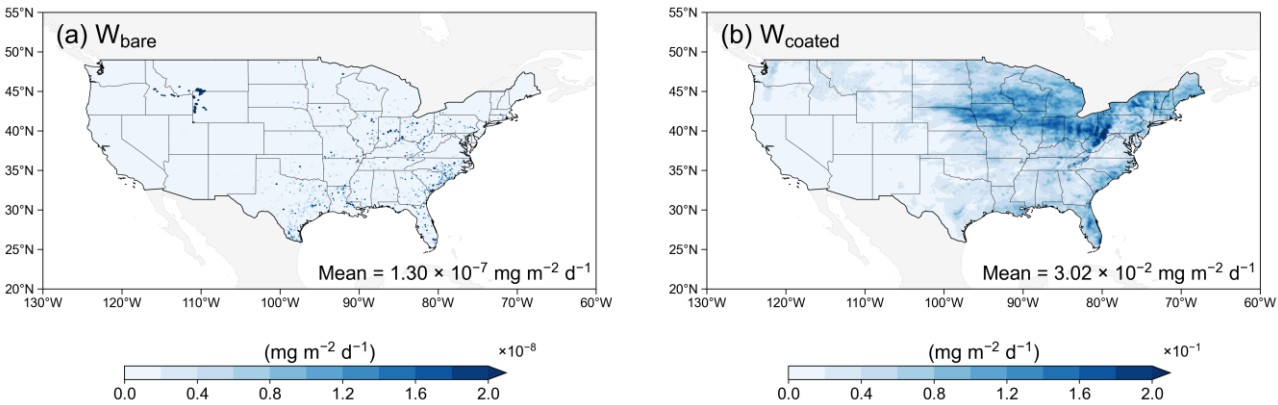

**Figure 9: Wet deposition of (a) Bare BC and (b) Coated BC simulated by the WRF-CMAQ-BCG model, with average hourly values for June.**

For the overall wet deposition of BC aerosol, the spatial distribution simulated by the WRF-CMAQ-BCG model is consistent with that simulated by the WRF-CMAQ model, mainly concentrated in the northeastern CONUS domain (Fig. 10). The average BC wet deposition simulated by the WRF-CMAQ-BCG model in the U.S. is $3.02 \times 10^{-2}$ mg m$^{-2}$ d$^{-1}$, which is approximately 17.7% less than the $3.67 \times 10^{-2}$ mg m$^{-2}$ d$^{-1}$ simulated by the WRF-CMAQ model. This reduction is because the WRF-CMAQ-BCG model categorized a portion of BC as hydrophobic Bare BC, which was not involved in nucleation scavenging, leading to a decrease in BC wet deposition. The 17.7% reduction highlights the significant impact of BC mixing state.





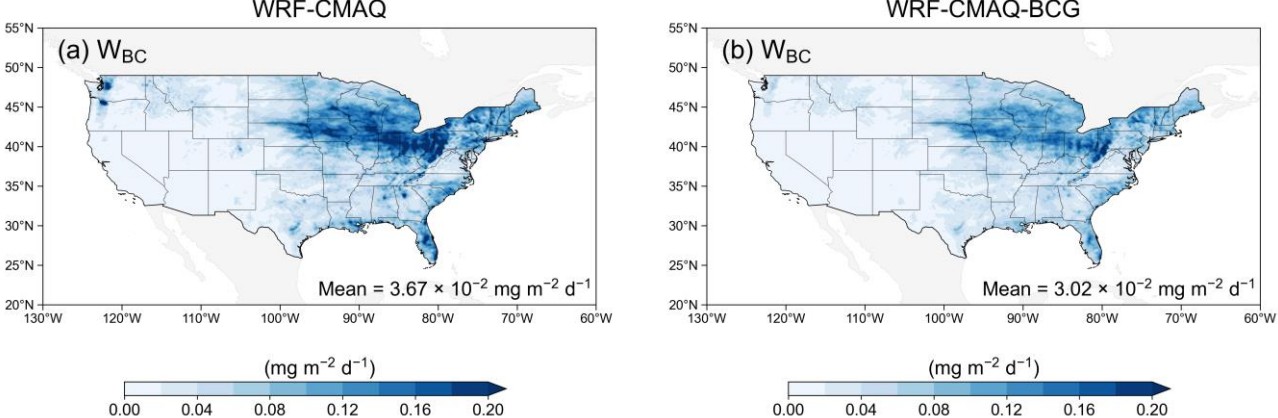

**Figure 10: BC wet deposition simulated by (a) the WRF-CMAQ model and (b) the WRF-CMAQ-BCG model, with average hourly values for June.**

345

The reduction in BC wet deposition may further alter the concentration of BC aerosol in the atmosphere. To investigate this potential scenario, we compared the surface BC mass concentration and column concentration between the two models, WRF-CMAQ and WRF-CMAQ-BCG (Fig. 11). The spatial distribution of BC mass concentration simulated by both models is similar. The average surface BC mass concentration simulated by the original model is $7.48 \times 10^{-2}$ μg m$^{-3}$, while in the new

350    model, the average is $7.96 \times 10^{-2}$ μg m$^{-3}$, indicating an increase in BC mass concentration. Similarly, the BC column concentration increases from an average of 0.19 mg m$^{-2}$ in the WRF-CMAQ model to 0.21 mg m$^{-2}$ in the WRF-CMAQ-BCG model, with an increase of 10.5%. This indicates that the fully internally mixed assumption, which does not account for the aging process, overestimates BC wet deposition and thus underestimates BC mass concentration.



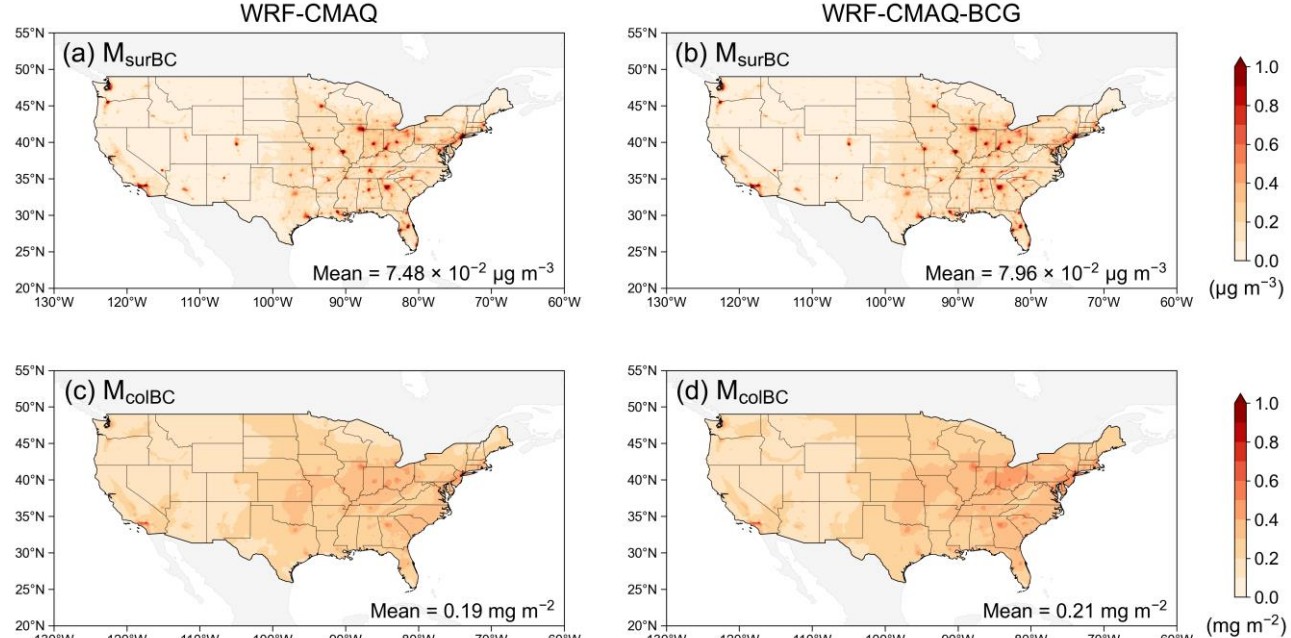

**Figure 11: Spatial distribution of (a, b) surface BC mass concentration and (c, d) column BC mass concentration simulated by (a, b) the WRF-CMAQ model and (c, d) the WRF-CMAQ-BCG model, with average hourly values for June.**

**4.4 Optical properties**

The aging process profoundly influences the optical properties of BC, in particular, the light-absorbing aspect. This study analyzed two key optical parameters at the 532 nm wavelength: the absorption coefficient ($b_{abs}$) and BC mass absorption cross-section ($MAC$), to investigate the changes in BC optical properties before and after accounting for the aging process (Yuan et al., 2021). The MAC values can be calculated as:

$$\text{MAC} = \frac{b_{abs}}{M_{BC}}, \tag{5}$$

where $M_{BC}$ represents BC mass concentration.

We compared the $b_{abs}$ and $M_{BC}$ values simulated by the WRF-CMAQ model and the WRF-CMAQ-BCG model, with the observed values at the T0 site (Fig. 12). For $b_{abs}$, the Mean Bias Error ($MBE$) value for the WRF-CMAQ model is $0.34 \times 10^{-5}$ m$^{-1}$, while for the WRF-CMAQ-BCG model is $0.19 \times 10^{-5}$ m$^{-1}$, representing a 44% reduction. The simulated $M_{BC}$ values for both models show little difference, with both $MBE$ values of 0.30 µg m$^{-3}$. The new model, which incorporated the aging process, demonstrates accuracy improvement in simulating $b_{abs}$ values with respect to observation data. However, the discrepancies in BC mass concentration ($M_{BC}$) between the two models are minimal, suggesting that variations in $M_{BC}$ play a minor role in the observed changes in the $MAC$ values.





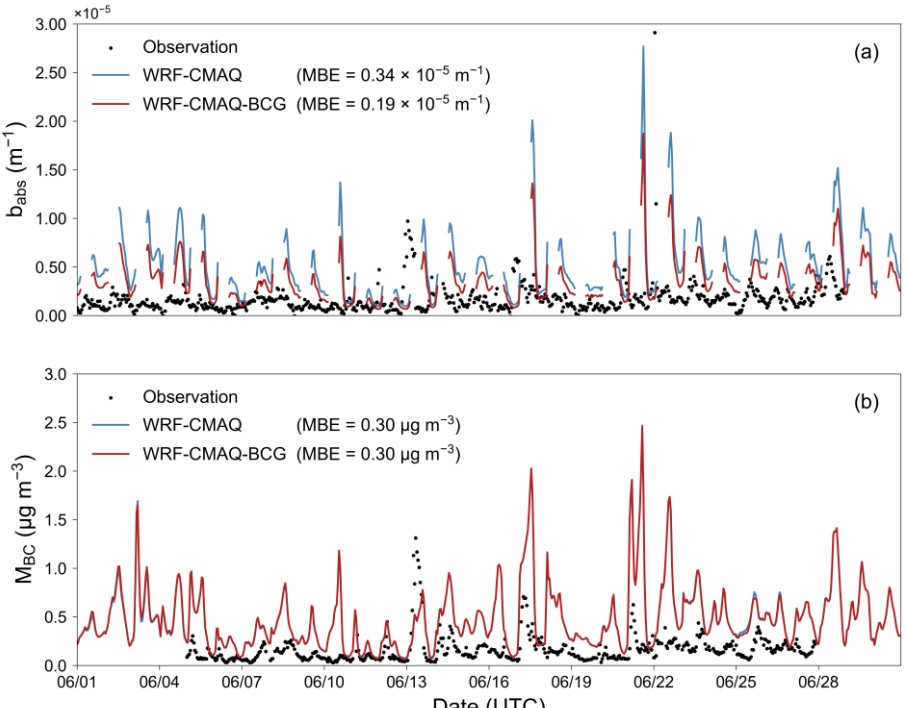

**Figure 12: Comparison of simulated and observed (a) absorption coefficient ($b_{abs}$) and (b) BC mass concentration ($M_{BC}$).**

375

We analyzed the performance of *MAC* values at the T0 site, with statistical data covering hourly observations and simulated results for June. As shown in Fig. 13, the median observed value of *MAC* is 9.75 m$^{-2}$ g $^{-1}$, while the median value from the original WRF-CMAQ model is 12.45 m$^{-2}$ g $^{-1}$, resulting in a difference of 2.7 m$^{-2}$ g $^{-1}$. In contrast, the median value from the WRF-CMAQ-BCG model is 8.77 m$^{-2}$ g $^{-1}$, which is closer to the observed median value. Furthermore, the broader sections

380 within each type of plot depict the primary distribution of *MAC* values, which is the most frequently occurring value. It is evident that the WRF-CMAQ-BCG model results are closer to the observations than the WRF-CMAQ model. We conclude that the WRF-CMAQ-BCG model simulates the optical properties of BC aerosol in a more precise manner.



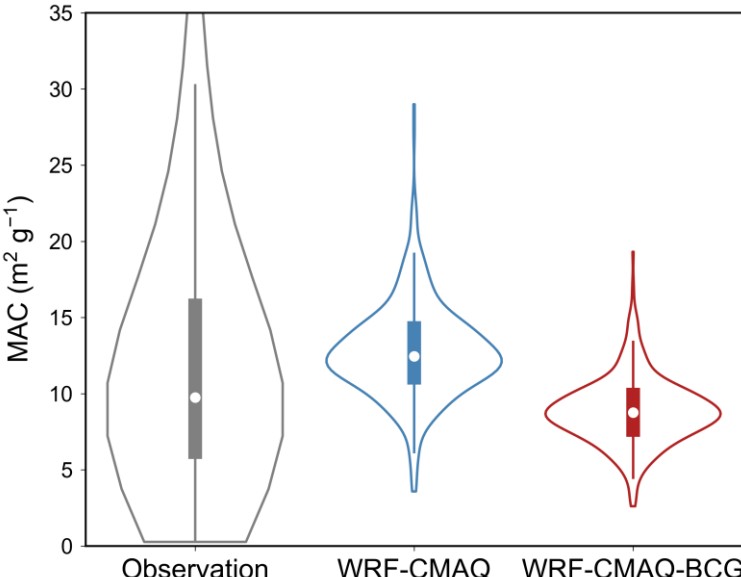

**Figure 13: Comparison of simulated and observed BC mass absorption cross-section (*MAC*). The width of the violin plot reflects the distribution of the *MAC* values, while the rectangular shapes inside represent miniature box-and-whisker plots. The white dots denote the median values of the results for that category.**

## 5 Conclusions

In this study, we developed the WRF-CMAQ-BCG model by integrating the BC aging process on top of the WRF-CMAQ model. This advancement involves the introduction of two distinct species, Bare BC and Coated BC, to characterize BC aging states and the development of a dedicated BC aging module. The aging process is represented by the conversion from Bare BC to Coated BC. Furthermore, we modified the cloud chemistry module and aerosol optical module to investigate the effects of changes in hydrophobicity and light absorption properties described in BC aging process.

Based on the WRF-CMAQ two-way coupled model, we have conducted a simulation over the US CONUS domain for June 2010 and have evaluated the results against observations from the CARES campaign at the T0 site located in Sacramento, California. The results indicate that the WRF-CMAQ-BCG model can provide reasonably accurate meteorological and chemical conditions for the BC aging process without compromising the computational accuracy of the original model. The results of our new module show the spatiotemporal variations of aging-related variables. The average aging rate is $2.26 \times 10^{-5}$ $s^{-1}$ and the average aging timescale is 17.49 h. Furthermore, the aging process allows BC aerosol mixing states to exhibit certain temporal and spatial distribution characteristics. Spatial analysis of the WRF-CMAQ-BCG model outputs demonstrates a clear aging behaviour: regions near emission sources exhibit a higher proportion of Bare BC, whereas areas farther from these sources show a dominance of Coated BC. The average number fraction of Coated BC is approximately 57%. For temporal variation characteristic, the number fraction of Coated BC simulated shows a distinct increase during the daytime period, with the peak occurring in the afternoon.



BC aerosol undergoes a conversion in its hydrophobic properties during the aging process, significantly impacting in-cloud scavenging mechanisms. Model simulations reveal distinct deposition patterns for different BC aging states: Bare BC exhibits a localized, point-like wet deposition pattern, while Coated BC displays a more widespread, zonal pattern. Notably, Coated BC emerges as the primary contributor to overall BC aerosol wet deposition. In addition, the average BC wet deposition simulated by the WRF-CMAQ-BCG model in the US region is $3.02 \times 10^{-2}$ mg m$^{-2}$ d$^{-1}$, which is 17.7% less than the $3.67 \times 10^{-2}$ mg m$^{-2}$ d$^{-1}$ simulated by the WRF-CMAQ model. The reduction in BC wet deposition makes the surface BC mass concentration and BC column concentration both higher. The average BC column concentration of 0.19 mg m$^{-2}$ in the WRF-CMAQ model increase to 0.21 mg m$^{-2}$ in the WRF-CMAQ-BCG model, representing an increase of 10.5%. This indicates that the fully internally mixed assumption in WRF-CMAQ model, which does not account for the aging process, overestimates BC wet deposition and thus underestimates BC mass concentration.

The new model's ability to simulate the coexistence of externally mixed Bare BC and internally mixed Coated BC, providing a better representation of atmospheric conditions, has prompted BC optical properties modification in the model. We compared the $b_{abs}$ values calculated by both the WRF-CMAQ model and the WRF-CMAQ-BCG model, with the observed values, their *MBE* values are $0.34 \times 10^{-5}$ m$^{-1}$ and $0.19 \times 10^{-5}$ m$^{-1}$, respectively. Both the median and mode values derived from the new model showed better agreement with observations compared to the original model's outputs. In conclusion, the WRF-CMAQ-BCG model, which accounts for BC aging process, enhances the capability to analyze aging-related variables and BC mixing state, as well as improves performance in terms of wet deposition and optical properties.

**Code availability**

The CMAQ model source codes are publicly available online at https://www.epa.gov/cmaq/access-cmaq-source-code. The WRF-CMAQ-BCG model source codes and other codes for this study are available on Zenodo at 425 https://zenodo.org/doi/10.5281/zenodo.12798673 (Jin et al., 2024a).

**Data availability**

The observation data provided by the Carbonaceous Aerosol and Radiative Effects Study (CARES) campaign are publicly available online at https://adc.arm.gov/discovery/#/results/s::CARES. Our analytic data that provide the major results are available on Zenodo at https://zenodo.org/doi/10.5281/zenodo.12798177 (Jin et al., 2024b).

**Author contribution**

Conceptualization: Jiandong Wang, Chao Liu; Data curation: Yuzhi Jin, David C. Wong, Golam Sarwar, Kathleen M. Fahey; Formal analysis: Jiandong Wang, Jiaping Wang, Jing Cai, Chao Liu; Funding acquisition: Jiandong Wang; Investigation:



Jiaping Wang, Zeyuan Tian, Shang Wu, Zhouyang Zhang; Methodology: Yuzhi Jin, Jiandong Wang, David C. Wong, Golam Sarwar, Kathleen M. Fahey, Chao Liu; Visualization: Yuzhi Jin; Writing – original draft: Yuzhi Jin; Writing –review & editing:
Yuzhi Jin, Jiandong Wang, David C. Wong, Chao Liu, Golam Sarwar, Kathleen M. Fahey, Shang Wu, Jiaping Wang, Jing Cai, Zeyuan Tian, Zhouyang Zhang, Jia Xing, Shuxiao Wang, Aijun Ding.

**Competing interests**

The authors declare that they have no conflict of interest.

**Disclaimer**

The views expressed in this paper are those of the authors and do not necessarily reflect the views or policies of the U.S. EPA.

**Acknowledgements**

This work was supported by the National Natural Science Foundation of China (42075098, Jiandong Wang) and the National Key R&D Program of China (2022YFC3701000, Task 5, Jiandong Wang). The model simulation is conducted in the High Performance Computing Center of Nanjing University of Information Science & Technology. We acknowledge the
observation data provided by the Carbonaceous Aerosol and Radiative Effects Study (CARES) campaign of the Atmospheric Radiation Measurement (ARM) program by the US Department of Energy (DOE).

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
