# Peer review of "Accounting for Black Carbon Aging Process in a Two-way Coupled Meteorology - Air Quality Model"

_EGUsphere, 2024_

## Author Comment (AC1)

***Response to the comments of Anonymous Referee #1 (egusphere-2024-2372)***

*The authors implemented a BC aging scheme into the WRF-CMAQ model and found that accounting for BC aging process improves simulated BC optical properties. They also found that adding the aging process significantly affect BC concentration distribution and wet deposition. Overall, the manuscript is well organized and the study fits well into the journal scope. However, there are a few places that require further descriptions and clarifications. I have a few comments and suggestions below for the authors to consider.*

**Response:** Thank you for your thoughtful and constructive feedback. We are grateful for your recognition of our work's organization and relevance to the journal's scope. Below, we address each of the comments and suggestions in detail. We have implemented all suggestions for improving our manuscript. Please find our point-by-point responses listed below. The reviewer's comments are in *Italic* followed by our responses and revisions (in blue). The modifications in the manuscript are highlighted in red.

*Major comments:*

*The major concern I have is the insufficient model evaluation. Currently, the authors only evaluated the model simulation at one measurement sites, while all their modeling analysis and conclusions come from regional results. Thus, regional-scale evaluation is needed, such as evaluations using IMPROVE aerosol measurement network, AERONET AOD/AAOD measurement network, EPA AirNow network, and/or MODIS AOD data.*

**Response:** Thanks for your suggestion. We have conducted a regional-scale evaluation of BC mass concentration ($M_{BC}$), $O_3$, $SO_2$, other gas concentration, and Aerosol Optical Depth (AOD). The data sources and evaluation results have been included in the newly added Supplementary Material, and a relevant explanation has been added to the main text.

"Figure 3 illustrates the spatial distribution of BC emission on the simulation domain in red and the blue star represents the ground observation station T0 of the Carbonaceous Aerosols and Radiative Effects Study (CARES) campaign, located in Sacramento, California. The observation dataset is available from the Atmospheric Radiation Measurement (ARM) program of the US Department of Energy (DOE) (https://adc.arm.gov/discovery/#/results/s::CARES). This comprehensive campaign collected a diverse array of data, encompassing aerosols, atmospheric conditions, cloud properties, and radiation data. The accuracy of the dataset has been widely recognized (Zaveri et al., 2012; Cahill et al., 2012). Data collected from this site is used as the benchmark for site evaluation and comparison, and the data used in our study are presented in Table 3. The data sources for the model regional-scale evaluation and analysis of results are provided in the supplementary material."

"To evaluate accuracy of the models' regional-scale results, we compared several common variables, as shown in Table S1. These include BC mass concentration ($M_{BC}$) data from the Interagency Monitoring of Protected Visual Environments (IMPROVE) measurement network, $O_3$, $SO_2$, and other gas concentration data from the Air Quality System (AQS) surface dataset, and Aerosol Optical Depth (AOD) data obtained from the Polarization and Directionality of the Earth's Reflectances (POLDER) satellite observations."

**Table S1. Performance Evaluation of WRF-CMAQ and WRF-CMAQ-BCG Models.**

| Results / Variables | Observation Mean | WRF-CMAQ Mean | MBE | RMSE | NMB | WRF-CMAQ-BCG Mean | MBE | RMSE | NMB |
|---|---|---|---|---|---|---|---|---|---|
| $M_{BC}$ (µg m⁻³) | 0.199 | 0.117 | -0.082 | 0.139 | -0.412 | 0.123 | -0.077 | 0.137 | -0.382 |
| $O_3$ (ppb) | 33.391 | 35.048 | 1.657 | 6.370 | 0.050 | 35.063 | 1.672 | 6.367 | 0.050 |
| $SO_2$ (ppb) | 1.934 | 1.740 | -0.194 | 4.202 | -0.100 | 1.740 | -0.194 | 4.202 | -0.100 |
| $NO$ (ppb) | 1.910 | 1.952 | 0.042 | 2.573 | 0.022 | 1.947 | 0.037 | 2.573 | 0.019 |
| $NO_2$ (ppb) | 6.539 | 7.795 | 1.256 | 4.290 | 0.192 | 7.787 | 1.248 | 4.277 | 0.191 |
| $AOD_{533}$ | 0.143 | 0.0586 | -0.0844 | 0.146 | -0.590 | 0.0585 | -0.0845 | 0.146 | -0.591 |

"With the Mean, Mean Bias Error (*MBE*), Root Mean Square Error (*RMSE*), and Normalized Mean Bias (*NMB*) metrics presented in Table S1, it can be seen that both the WRF-CMAQ and WRF-CMAQ-BCG models simulated accurately. In addition, the mean error of $M_{BC}$ simulated by the new model is -0.077 µg m⁻³, which is closer to the observations compared to the original model's values of -0.082 µg m⁻³, indicating a slight improvement in alignment with the observation (Fig. S1). Overall, the new model, while adding new functionalities and enhancing the accuracy of optical calculations, does not compromise the simulation of other variables and slightly improves the simulation of BC mass concentration."

[Figure]

**Figure S1: Comparison of BC mass concentration ($M_{BC}$) errors between simulations and observations: (a) the WRF-CMAQ model, (b) the WRF-CMAQ-BCG model.**

***Specific comments:***

*1) Lines 87-88: This "limited impact" is not accurate, since different BC mixing states affect aerosol hygroscopicity and hence wet deposition, which subsequently change the mass concentration.*

**Response:** Thanks for your suggestion, and we apologize for the inaccurate statement. We have revised it to provide a clearer explanation.

"The Community Multiscale Air Quality (CMAQ) model, developed by the US Environmental Protection Agency (EPA), is widely used in the research community as well as in the US government as a regulatory model, and it continues to evolve. To account for the interactive two-way feedback between aerosols and meteorological conditions, Wong et al. (2013) developed the Weather Research and Forecasting - Community Multiscale Air Quality (WRF-CMAQ) two-way coupled model. The mixing state of BC aerosol is simplified to a fully internally mixed state in the CMAQ model. When considering the influence of aerosols on meteorology, this simplification has a more pronounced impact. Therefore, incorporating the BC aging process in the WRF-CMAQ model is essential, as it significantly influences hydrophobicity and light absorption."

*2) Line 103: "... coated with scattering aerosol component". This is not very accurate since BC can also be coated by some absorbing organics.*

**Response:** Thank you for your suggestion. We have changed "scattering aerosol component" to "other aerosol component" and added the explanation "(some absorbing organics that coat BC are not considered in this study)" in the second paragraph of the introduction in the revised manuscript.

"The BC component exhibits very low chemical reactivity and is refractory (Bond et al., 2013). Consequently, previous studies have typically considered BC aerosol as chemically inert, primarily serving as a reaction interface for other chemical reactions due to its unique morphology (Monge et al., 2010). However, freshly emitted BC aerosol experiences condensation, coagulation, and heterogeneous oxidation processes during atmospheric transport, becoming coated by scattering aerosol components and converted into aged BC aerosol (some absorbing organics that coat BC are not considered in this study)."

*3) Line 108: Even for two-way coupled WRF-CMAQ model, there is no aerosol indirect effect considered? Did the authors mean the standard EPA-version of WRF-CMAQ? There might be a WRF-CMAQ version from individual research group that may already have this capability. Please double check.*

**Response:** Thanks for your suggestion. Yes, we are referring to "the standard EPA-version of WRF-CMAQ." While Yu et al. (2014) studied the indirect effects in WRF-CMAQ, these have not been incorporated into the standard version of the model. This is also an area of ongoing work for us. We have revised the sentence in the manuscript as: "(the EPA's publicly released WRF-CMAQ model does not yet have the indirect radiative effect capability)".

*4) In Figures 1 and 2, the authors showed that the model includes aerosol-cloud interaction, but in the description of Section 2.1, it seems that the model does not account for the aerosol indirect effect. This needs further clarification.*

**Response:** Thanks for your comment. We emphasized the impact of clouds on BC aerosol in the figures, particularly the wet deposition caused by in-cloud scavenging. The role of BC aerosol as CCN is indeed not considered in CMAQ. We apologize for any confusion this may have caused and have updated Figures 1 and 2 accordingly (Figure 1 and Table 2).

[Figure]

**Figure 1:** **The BC mixing state in the WRF-CMAQ model and the WRF-CMAQ-BCG model.**

**Table 2. Comparison of BC aerosol in major processes (Aitken and accumulation modes).**

| Processes \ Species | | BC (Original) | Bare BC (New) | Coated BC (New) |
|---|---|---|---|---|
| | Emission | Yes | Yes | No |
| | BC Aging | No Aging process | Bare BC is aged to Coated BC | |
| Wet Deposition | Impact scavenging | Yes | Yes | Yes |
| | Nucleation scavenging | Yes | No | Yes |
| | Aerosol Optics | Core-shell sphere | Homogeneous sphere | Core-shell sphere |

*5) Equation (3): Which term is for condensation (fast-aging)? It seems that the first beta\*[OH] term represents chemical aging and alpha represents coagulation.*

**Response:** Thanks for your comment. The "$\beta$" is for condensation (fast-aging), we have updated the description of the equation in the revised manuscript.

"$\beta$ is estimated by assuming an e-folding aging timescale of 2.5 days for condensation, and $\alpha$ is estimated by assuming a 20 days e-folding lifetime for coagulation (Liu et al., 2011; Huang et al., 2013; Oshima and Koike, 2013)."

*6) Section 2.3: the use of "cloud chemistry" is confusing since BC does not undergo any chemical process in the cloud droplets in the model. Also, the description of this part is not very clear. How did the authors*

*set the hydrophobicity of coated BC? What scheme did the authors use to compute CCN from aerosol number concentration and hygroscopicity? What equations did the authors use to compute the impact scavenging of BC aerosol? More details are needed.*

**Response:** Thanks for your suggestions. The term "cloud chemistry module" was used in the CMAQ Users Guide. To avoid confusion, it has been changed to "wet deposition module" in the revised manuscript.

We apologize for the ambiguous description that led to your confusion. We defined "Coated BC" species as hydrophilic and "Bare BC" species as hydrophobic. When these two new species enter the wet deposition module, we modified the primary in-cloud scavenging algorithm based on their differences in hydrophobicity. Since the WRF-CMAQ model does not yet account for aerosol indirect effects, CCN is calculated based on assumptions, and removal rates are computed using cloud water content and precipitation rate. BC in the accumulation mode is removed by nucleation scavenging, while BC in the Aitken mode is treated as interstitial aerosols subjected to impact scavenging (Binkowski and Roselle, 2003). In the CMAQ model, the method for calculating impact scavenging is derived from Binkowski and Roselle (2003), as shown below:

$$E_s = 2\pi \, m_{1c} \, \langle D \rangle \, (1 + 0.5 \, Pe^{1/3})$$

where $E_s$ is the scavenging efficiency, $m_{1c}$ is the first moment of the cloud droplet distribution, $\langle D \rangle$ is the polydisperse diffusivity, and $Pe$ is Peclet number.

Within the framework of the WRF-CMAQ model, we modified the wet deposition algorithm to ensure that hydrophobic Bare BC in the accumulation mode is not removed by nucleation scavenging, while Aitken-mode Bare BC undergoes impact scavenging. After impact scavenging, Bare BC is converted to Coated BC due to water envelopment, which continues its removal from the atmosphere. The above additional description has been added in Section 2.3 in the revision.

"In the WRF-CMAQ model, removal rates are computed using cloud water content and precipitation rate. The original model further differentiates scavenging mechanisms based on particle size: BC aerosol in the accumulation mode undergoes nucleation scavenging, while BC aerosol in the Aitken mode experiences impact scavenging as interstitial aerosol (Binkowski and Roselle, 2003). The hydrophobicity changes caused by BC aging process mainly affect nucleation scavenging in the way that hydrophobic Bare BC cannot act as CCN, while hydrophilic Coated BC can. In other words, Bare BC cannot undergo nucleation scavenging. Within the framework of the WRF-CMAQ model, we modified the wet deposition algorithm to ensure that hydrophobic Bare BC in the accumulation mode is not removed by nucleation scavenging, while Aitken-mode Bare BC undergoes impact scavenging. After impact scavenging, Bare BC is converted to Coated BC due to water envelopment, which continues its removal from the atmosphere, as illustrated in right panel of Fig. 1. Overall, these updates in the wet deposition module in this work enhance the representation of BC aerosol in various aspects."

*7) Section 2.4: It is not clear that how the authors compute the number concentrations of bare and coated BC. (1) Are these number concentrations two new prognostic variables tracked by the model? What are the size distributions of bare and coated BC particles used during the mass-to-number conversion? More*

*clarifications are needed. (2) Another key uncertainty factor related to the calculation of BC optics is the particle structure. Many previous studies have shown that using core-shell assumption for coated BC and spherical shape for bare BC cannot realistically represent BC particle optics (e.g., https://doi.org/10.1029/2021GL096437; https://doi-org.cuucar.idm.oclc.org/10.1021/acs.estlett.7b00418; https://doi.org/10.5194/acp-15-11967-2015). It may be challenging to add the particle structure info into the model, but some discussions on this uncertainty factor will be helpful.*

**Response:** Thank you for your suggestions.

(1) Bare BC and coated BC are assumed to have the same size distribution, have the same density and volume but are merely in different aging states. Therefore, the number fraction of Coated BC can be calculated from the mass fraction of Coated BC, as shown in Eq.(4). When calculating aerosol optical properties, BC particle size information is highly sensitive. For aerosol optics calculation, the volumes of other species (as the shell) are added to the volume of Coated BC to recalculate the particle size. We have revised the manuscript to reflect this point of view.

"In our WRF-CMAQ-BCG model, we calculated the optics of Bare BC and Coated BC separately. We introduced a variable, the number fraction of Coated BC ($NF_{\text{coated}}$). The $NF_{\text{coated}}$ variable was brought into the aerosol optics module for translating BC core back to Bare BC and Coated BC. Only Coated BC can be a core surrounded by a shell. Once encapsulated, Coated BC becomes a BC-containing particle, represented as a core-shell sphere, and its particle size information is recalculated based on the volume of the Coated BC core and the shell. Its optical properties are calculated using Core-shell Mie theory. In contrast, Bare BC is represented as a homogeneous sphere, with the particle size recalculated using the volume of Bare BC, and its properties are calculated using the standard Mie theory. By apportioning the BC core, the overestimation of aerosol light absorption can be corrected.

$$NF_{\text{coated}} = \frac{N_{\text{coated}}}{N_{\text{bare}} + N_{\text{coated}}} = \frac{V_{\text{coated}}}{V_{\text{bare}} + V_{\text{coated}}} = \frac{M_{\text{coated}}}{M_{\text{bare}} + M_{\text{coated}}} , \qquad (4)$$

where $NF_{\text{coated}}$ represents the number fraction of Coated BC, $N_{\text{bare}}$ and $N_{\text{coated}}$ are the number concentration of Bare BC and Coated BC aerosol, respectively. $V_{\text{bare}}$ and $V_{\text{coated}}$ are the volume of Bare BC and Coated BC aerosol, respectively. $M_{\text{bare}}$ and $M_{\text{coated}}$ are the mass concentration of Bare BC and Coated BC aerosol, respectively. Bare BC and Coated BC are merely in different aging states, they are essentially BC aerosol with the same density and volume. Therefore, the number fraction of Coated BC can be calculated from the mass fraction of Coated BC. "

(2) Thank you for suggesting the addition of a discussion on particle structure. The CMAQ model currently cannot incorporate particle structure information and still uses the spherical shape assumption (Core-shell sphere and Homogeneous sphere), which may introduce some biases. We have included relevant discussion in the revised manuscript.

"In the WRF-CMAQ model, the light absorption of aerosols is entirely attributed to BC aerosol. BC aerosol is considered the core, with water-soluble aerosols, insoluble aerosols, aerosol water, and sea salt as the shell. Each substance has its corresponding refractive index across 14 wavelengths under the Rapid Radiative Transfer Model (RRTM) for global climate model (GCM) applications (RRTMG) scheme. For aerosols containing BC in the Aitken and accumulation modes, the Core-shell Mie theory is employed to calculate their optical characteristics (coarse mode aerosols without BC component are not considered in this study). The particle structure information cannot be fully represented in the current WRF-CMAQ model, as the spherical shape assumption is used for calculations. This simplification may introduce biases in the results (He et al., 2015; Wang et al., 2017, 2021). The potential impacts of such structural simplifications are not

**8)Lines 220-221: How sensitive the model results are to the assumption of equal fraction of bare and coated BC in the initial and boundary conditions?**

**Response:** Thank you for your comment. We used the entire month of May as the spin-up period. With such a sufficiently long spin-up time, the model results are not sensitive to variations in the fraction of bare and coated BC in the initial condition. Similarly, BC aerosol transported over long distances undergo significant aging, justifying the assumption of equal fraction of bare and coated BC in the boundary condition.

**9)Line 223: "the model assimilated data" Did the authors mean they also used data assimilation in their model simulations? Did the authors conduct 3 different simulations by using these 3 datasets (FNL, NAM, and NARR), respectively?**

**Response:** Thank you for your comment. The sentence has been corrected in the revised manuscript. We used the Weather Research and Forecasting Data Assimilation (WRFDA) method for data assimilation, with the assimilated data sourced from the North American Mesoscale Forecast System (NAM), provided by the National Weather Service's National Centers for Environmental Prediction (NCEP).

"The WRF model parameterization schemes selected for the simulation case are listed in Table 3. The CMAQ model employs the "cb6r5_AERO7" chemical mechanism and utilizes the Rosenbrock solver. Additionally, we used the Weather Research and Forecasting Data Assimilation (WRFDA) method for data assimilation, with the assimilated data sourced from the North American Mesoscale Forecast System (NAM), provided by the National Weather Service's National Centers for Environmental Prediction (NCEP)."

**10)Figures 4-5: It seems that including BC aging only has negligible benefits on model performance. How to better justify the need to include this aging scheme?**

**Response:** Thanks for your comment. The results in Figures 4-5 are intended to demonstrate that the new model, like the original model, can provide accurate meteorological and chemical conditions for considering the BC aging process in this study. We apologize for any confusion caused by our ambiguous explanation, and we have provided a clearer expression in the revised manuscript.

"To evaluate the accuracy of meteorological and chemical conditions for considering the BC aging process in this study, we compared the simulation results of the WRF-CMAQ model and the WRF-CMAQ-BCG model with various meteorological observations, as well as the volumetric concentrations of several gases at the T0 site in the CARES campaign. Figure 3 illustrates…Clearly, the inclusion of the BC aging process does not degrade the original model's accuracy in simulating these common variables and the models can provide reasonably accurate meteorological and chemical conditions for the aging process."

The inclusion of the BC aging process does not degrade the original model's accuracy in simulating some

common variables. Instead, it adds new functionalities to the WRF-CMAQ model, such as simulating BC mixing state distributions, tracking BC aging timescales, and separately analyzing the wet deposition of BC with different mixing states. Additionally, it enhances the model's performance in simulating BC aerosols, including their mass concentration and optical properties. These points illustrate the necessity of considering the BC aging process in the WRF-CMAQ model.

**References**

Binkowski, F.S., and Roselle, S.J.: Models-3 Community Multiscale Air Quality (CMAQ) model aerosol component. 1. Model description, J. Geophys. Res., 108, 4183, doi:10.1016/J. SCITOTENV.2017.06.082, 2003.

He, C., Liou, K. N., Takano, Y., Zhang, R., Levy Zamora, M., Yang, P., Li, Q., and Leung, L. R.: Variation of the radiative properties during black carbon aging: theoretical and experimental intercomparison, Atmos. Chem. Phys., 15(20), 11967-11980, doi:10.5194/acp-15-11967-2015, 2015.

Huang, Y., Wu, S., Dubey, M. K., and French, N. H. F.: Impact of aging mechanism on model simulated carbonaceous aerosols, Atmos. Chem. Phys., 13(13), 6329-6343, doi:10.5194/acp-13-6329-2013, 2013.

Liu, J., Fan, S., Horowitz, L. W., and Levy, H.: Evaluation of factors controlling long-range transport of black carbon to the Arctic, J. Geophys. Res. Atmos., 116(D4), doi:10.1029/2010JD015145, 2011.

Oshima, N., and Koike, M.: Development of a parameterization of black carbon aging for use in general circulation models, Geosci. Model Dev., 6(2), 263-282, doi:10.5194/gmd-6-263-2013, 2013.

Wang, Y., Liu, F., He, C., Bi, L., Cheng, T., Wang, Z., Zhang, H., Zhang, X., Shi, Z., and Li, W.: Fractal dimensions and mixing structures of soot particles during atmospheric processing, Environ. Sci. Technol. Lett., 4(11), 487-493, doi:10.1021/acs.estlett.7b00418, 2017

Wang, Y., Li, W., Huang, J., Liu L., Pang, Y., He, C., Liu, F., Liu, D., Bi, L., Zhang, X., and Shi Z.: Nonlinear Enhancement of Radiative Absorption by Black Carbon in Response to Particle Mixing Structure, Geophys. Res. Lett., 48(24), e2021GL096437, doi:10.1029/2021GL096437, 2021.

Yu, S., Mathur, R., Pleim, J., Wong, D., Gilliam, R., Alapaty, K., Zhao, C., and Liu, X.: Aerosol indirect effect on the grid-scale clouds in the two-way coupled WRF–CMAQ: model description, development, evaluation and regional analysis. Atmos. Chem. Phys., 14(20), 11247-11285, doi:10.5194/acp-14-11247-2014, 2016.

---

## Author Comment (AC2)

***Response to the comments of Anonymous Referee #2 (egusphere-2024-2372)***

*In this study, a BC aging parameterization was implemented into the WRF-CMAQ model. BC aging processes are important for accurately estimating the spatial distribution, atmospheric lifetime, optical properties, and activation to cloud particles of BC. While BC has been treated as internally mixed particles in CMAQ, the authors have classified BC into externally mixed and internally mixed particles and introduced an aging parameterization for converting externally mixed particles to internally mixed particles, as well as a scheme for calculating the differences in cloud activation and optical properties between externally mixed and internally mixed BC particles. This study contains interesting aspects as a BC modeling study. However, many similar studies have been published in the last 15 years, and this study lacks scientific novelty. Considering this point, I cannot recommend this study as an ACP paper.*

**Response:** Thank you so much for your thorough review and agreement on the importance of this study. We believe our manuscript is not only technically valuable but also scientifically novel for the following reasons:

1. This is the first time the BC aging process has been considered and tracked within the framework of the CMAQ model and WRF-CMAQ coupled model. The CMAQ model, a widely accepted chemistry transport model, recognized for regulatory applications in the US. No doubt BC aging is a worthwhile process (which the reviewer also noted), and this work acquired the BC aging in our focal model, WRF-CMAQ coupled model. Thus, our model development reflects a sophisticated and state-of-the-art integration of different physical processes. Although some of the parametrizations are adopted from previous works for other models, our work is path breaking in the following two aspects:

(1) We introduced a new scheme using two additional species: Bare BC and Coated BC, to represent different aging states. Significant differences in BC properties drove us to modify wet deposition and aerosol optics algorithms. Unlike other methods, which added new modes in modal models, in previous studies, our approach not only reduced overall complexity, but also enhanced the model's functionality and accuracy with minimal computational demand.

(2) Our in-cloud scavenging calculation method accounts for evolving hydrophilicity of BC aerosol, improving the accuracy of wet deposition simulations. Specifically, Bare BC, being hydrophobic, does not engage in nucleation scavenging but is subject to impact scavenging. After impact scavenging, Bare BC becomes hydrophilic and is subsequently removed as Coated BC.

2. Besides the model development itself, this manuscript showed some new findings and conclusions as well. We are regretted that we didn't present them clearly in the original submission, and have improved them in the revision.

(1) Previous studies have primarily focused on the effects of the BC aging process on mixing state and optical properties, with less emphasis on its impact on concentration. In this work, we found that the

changes in wet deposition caused by BC aging process can significantly impact BC concentrations. In the US CONUS domain, considering the aging process results in a 17.7% reduction in simulated wet deposition, accompanied by a 10.5% increase in BC column concentration.

(2) We have explicitly provided the spatial and temporal distribution characteristics of BC aerosol mixing states: Bare BC is prevalent near emission sources, while Coated BC is more common farther from sources, and the Number Fraction of Coated BC increases during the daytime.

(3) The influence of BC aging on its optical properties has been a long-standing uncertainty, although with increasingly detailed characterizations of BC mixing states (Wang et al., 2018; Chen et al., 2023). The CMAQ model, as a well-established air quality model, accounts for over a hundred aerosol species and hundreds of chemical reactions, enabling more accurate aerosol information. Within the WRF-CMAQ model framework, we found that incorporating differences in BC mixing states due to aging alone improves the alignment of simulation results with observational data. The median MAC (mass absorption cross-section) simulated by our new model is 8.77 m²/g, approximately 30% lower than the original model's value of 12.45 m²/g, bringing it closer to the observed value of 9.75 m²/g. This result was achieved without accounting for aerosols scattering separately. This implies in models with comprehensive physical-chemical processes, optical calculation process could be simplified in future.

Thus, we do believe our work offers both technical contributions and scientific advancements, and think it is suitable for both ACP and specialized journals on model development. We are considering ACP due to its high impact and profound influence within the communities that focused on both model development and scientific researches.

Please find our point-by-point responses listed below. The reviewer's comments are in *Italic* followed by our responses and revisions (in blue). The modifications in the manuscript are highlighted in red.

***Major comments:***

*The introduction is generally well structured and covers the important previous studies. However, the model developed in this study has already been developed and used in the papers listed in the introduction, indicating that this study lacks scientific originality. It can be said that the optical property part is relatively new, but there have already been many studies focusing on the differences in light absorption efficiency caused by the mixing state of BC particles.*

*Is it considered that this research is not new scientifically, but new to the CMAQ model? Or does it contain some new scientific findings? I think this research falls into the former category. In that case, this study is not appropriate for ACP. I strongly recommend that the authors submit this study to another model development journal (e.g., GMD).*

**Response:** Thank you so much for such a candid comment. Through collaboration with the US EPA CMAQ development team, this is the first time that BC aging process has been considered and tracked within the framework of the CMAQ and WRF-CMAQ models. As discussed in our previous response, our work contributes significantly to both model development and scientific understanding, and not only the optical properties are new. Given the comprehensive nature of this work, we believe it provides valuable advancements beyond previous studies. Therefore, we believe that ACP is an appropriate journal for this publication.

*2) Related to the comment above, the BC aging parameterization used in this study is the same as that developed and used in previous studies, and there is nothing new about it. In addition, the methods section needs substantial revision because there are many things that are not adequately described. For example, does the model treat aerosols that do not contain BC (BC-free particles) in the accumulation mode? Considering BC-free particles is important for estimating the mixing state and optical properties of BC, but it is not clear from the text how the model distinguishes non-BC species between BC-containing particles (used for coating) and BC-free particles (see comment 4 below).*

**Response:** Thank you for your comments. As mentioned in my previous response, our approach to incorporating the BC aging process differs from the methods summarized in the third paragraph of the introduction regarding previous models.

We also appreciate your suggestion to expand the methods section. We apologize for not providing a more detailed explanation earlier and have revised the manuscript accordingly (in red) to strengthen the methodology description.

(1) In Section 2.1, "New Model", we added a new Table 2 (please see comment 4 below) and an explanation highlighting the differences in the treatment of BC aerosol between the original model and the new model.

"In the original model, BC aerosol did not include the aging process and was treated as being in a completely internally mixed state. This approach neglected the presence and impact of externally mixed BC aerosol in the real atmosphere, leading to inaccuracy in the numerical simulation of BC aerosol. In our new model, the differences of BC aerosol across major processes under different aging states are considered. Freshly emitted BC aerosol (Bare BC) enters the BC aging module from the emission module and gradually converts into Coated BC through the aging process. When Bare BC and Coated BC enter the wet deposition module, hydrophobic Bare BC cannot undergo nucleation scavenging during in-cloud scavenging, in turns affecting the wet deposition of BC aerosol. Bare BC exists in an external mixing state, whereas Coated BC has stronger light absorption due to the lensing effect. Therefore, Bare BC and Coated BC need to be considered separately when calculating aerosol optics."

(2) In Section 2.2, "BC Aging Module", we revised the description of the method for the virtual chemical reaction.

"Based on the selected OH aging scheme, the dynamic process of BC aging is represented by setting a virtual reaction, wherein Bare BC progressively converts into Coated BC. The aging rate is used as the reaction rate for this virtual chemical reaction."

(3) In Section 2.3, "Cloud Chemistry Module", we included a clarification that this study does not consider precipitation scavenging and focuses mainly on in-cloud scavenging processes.

"In the WRF-CMAQ model, the wet deposition module carries the following tasks: calculates in-cloud scavenging and precipitation scavenging of BC aerosol, performs aqueous chemistry calculations, and accumulates wet deposition. Precipitation scavenging associated with precipitation forms such as rain, snow, and graupel, and is minimally affected by changes in BC aerosol properties before and after aging. Therefore, this study does not consider the differences in precipitation scavenging process. In contrast, in-cloud scavenging (includes nucleation scavenging and impact scavenging) is strongly associated with BC aging process, reflecting the hydrophobicity differences between Bare BC and Coated BC."

(4) In Section 2.4, "Aerosol Optics Module", we revised potentially misleading description related to coarse mode scattering aerosols and clarified the explanation of the aerosol optics algorithm modification.

"For aerosols containing BC in the Aitken and accumulation modes, the Core-shell Mie theory is employed to calculate their optical characteristics (coarse mode aerosols without BC component are not considered in this study). The particle structure information cannot be fully represented in the current WRF-CMAQ model, as the spherical shape assumption is used for calculations. This simplification may introduce biases in the results (He et al., 2015; Wang et al., 2017, 2021). The potential impacts of such structural simplifications are not addressed in this study. In our WRF-CMAQ-BCG model, we calculated the optics of Bare BC and Coated BC separately. We introduced a variable, the number fraction of Coated BC ($NF_{coated}$). The $NF_{coated}$ variable was brought into the aerosol optics module for translating BC core back to Bare BC and Coated BC. Only Coated BC can be a core surrounded by a shell. Once encapsulated, Coated BC becomes a BC-containing particle, represented as a core-shell sphere, and its particle size information is recalculated based on the volume of the Coated BC core and the shell. Its optical properties are calculated using Core-shell Mie theory. In contrast, Bare BC is represented as a homogeneous sphere, with the particle size recalculated using the volume of Bare BC, and its optics properties are calculated using standard Mie theory. By apportioning the BC core, the overestimation of aerosol light absorption can be corrected."

Regarding your comment on BC-free particles in the accumulation mode, the current version of the model does not treat these particles (for specific details, please refer to comment 4 below).

*Specific comments:*

*3) L41: formation or altering -> formation and altering.*

**Response:** Thanks for your suggestion. We have changed "formation or altering" to "formation and altering" in revised manuscript.

*4) L144: Figure 2: Are BC-free particles considered in the accumulation mode? In Figure 2, it appears that BC-free particles are treated as scattering aerosols. If so, how are scattering aerosols in coated BC and scattering aerosols (BC-free particles) treated in the model? Are they treated as separate aerosol variables?*

**Response:** Thanks for your comments. We did not consider BC-free particles in the accumulation mode in our model. Apologies for any confusion caused. The WRF-CMAQ model operates under the assumption of complete internal mixing, and calculating aerosol optics on a modal basis. The Aitken mode and accumulation mode containing BC aerosol both utilize the Core-shell Mie method for optical calculations, while coarse mode aerosols that lack BC content are treated as BC-free particles and calculated using the standard Mie method. In this study, to account for the effects of the BC aging process, we specifically considered the Aitken and accumulation modes with BC, with a distinguishing mixing state of BC aerosol without including BC-free particles. We have replaced Figure 2 with Table 2 and revised the text in Section 2.4 to clarify this point.

**Table 2. Comparison of BC aerosol in major processes (Aitken and accumulation modes).**

| Processes \ Species | | BC (Original) | Bare BC (New) | Coated BC (New) |
|---|---|---|---|---|
| | **Emission** | Yes | Yes | No |
| | **BC Aging** | No Aging process | Bare BC is aged to Coated BC | |
| **Wet Deposition** | Impact scavenging | Yes | Yes | Yes |
| | Nucleation scavenging | Yes | No | Yes |
| | **Aerosol Optics** | Core-shell sphere | Homogeneous sphere | Core-shell sphere |

"In the WRF-CMAQ model, the light absorption of aerosols is entirely attributed to BC aerosol. BC aerosol is considered the core, with water-soluble aerosols, insoluble aerosols, aerosol water, and sea salt as the shell. … For aerosols containing BC in the Aitken and accumulation modes, the Core-shell Mie theory is employed to calculate their optical characteristics (coarse mode aerosols without BC component are not considered in this study). The particle structure information cannot be fully represented in the current WRF-CMAQ model, as the spherical shape assumption is used for calculations. This simplification may introduce biases in the results (He et al., 2015; Wang et al., 2017, 2021). The potential impacts of such structural simplifications are not addressed in this study. In our WRF-CMAQ-BCG model, we calculated the optics of Bare BC and Coated BC separately. We introduced a variable, the number fraction of Coated BC ($NF_{coated}$). The $NF_{coated}$ variable was brought into the aerosol optics module for translating BC core back to Bare BC and Coated BC. Only Coated BC can be a core surrounded by a shell. Once encapsulated, Coated BC becomes a BC-containing particle, represented as a core-shell sphere, and its particle size information is recalculated based on the volume of the Coated BC core and the shell. Its optical properties

are calculated using Core-shell Mie theory. In contrast, Bare BC is represented as a homogeneous sphere, with the particle size recalculated using the volume of Bare BC, and its optics properties are calculated using standard Mie theory. By apportioning the BC core, the overestimation of aerosol light absorption can be corrected."

*5) L167-168: It is not correct to assume that the aging speed of coagulation is constant. Coagulation occurs very fast near sources and has a large contribution to BC aging. The speed of coagulation aging is highly dependent on aerosol concentrations.*

**Response:** Thank you for your suggestion. We agree that coagulation can occur rapidly near sources. However, coagulation primarily manifests as aggregation between BC particles forming chain-like structures (Bond et al., 2013). Collisions between externally mixed black carbon particles do not significantly alter their mixing state. Additionally, we ran CMAQ which is a regional air quality model, with a spatial resolution of 12 km, therefore, it was not able capturing rapid coagulation that took place close to emission sources. In our simulation, the daily average concentration of $PM_{2.5}$ reached a maximum value of 33.48 μg m$^{-3}$. Within this range, Coagulation does not play a dominant role compared to Condensation (particularly pronounced in the accumulation mode, where BC predominantly present). Based on these considerations, we have followed other studies (Liu et al., 2011; Huang et al., 2013; Oshima and Koike, 2013) to employ a constant coagulation rate.

*6) L178: What are the particle size distributions of Bare BC and Coated BC? They should have different dry deposition speed because their particle size distributions are different due to the coating species and water uptake.*

**Response:** Thank you for your comment. In this study, Bare BC and Coated BC are assumed to have the same particle size distribution because both consist of elemental carbon (EC) but have different aging properties. When calculating aerosol optical properties, the volumes of other species (as shell) are added to the volume of Coated BC for recalculating the particle size. We have added a relevant explanation in Section 2.4 in the revised manuscript as follows.

"In our WRF-CMAQ-BCG model, we calculated the optics of Bare BC and Coated BC separately. We introduced a variable, the number fraction of Coated BC ($NF_{coated}$). The $NF_{coated}$ variable was brought into the aerosol optics module for translating BC core back to Bare BC and Coated BC. Only Coated BC can be a core surrounded by a shell. Once encapsulated, Coated BC becomes a BC-containing particle, represented as a core-shell sphere, and its particle size information is recalculated based on the volume of the Coated BC core and the shell. Its optical properties are calculated using Core-shell Mie theory. In contrast, Bare BC is represented as a homogeneous sphere, with the particle size recalculated using the volume of Bare BC, and its optics properties are calculated using standard Mie theory. By apportioning the BC core, the overestimation of aerosol light absorption can be corrected."

Regarding dry deposition, as mentioned in Section 2.3, primary influencing factors are meteorological conditions and land surface types. This implies there is only a minor difference in the dry deposition rates of Bare BC and Coated BC. As shown in Emerson et al. (2020), the dry deposition velocities for both Bare BC and Coated BC are approximately 0.2 cm s$^{-1}$, indicating minimal differences. Moreover, research

by Textor et al. (2006) indicates that wet deposition accounts for 79% of the total removal of BC aerosol. Therefore, we mainly emphasized the wet deposition, while the dry deposition settings remain consistent with the original model.

[Figure]

(Figure in Emerson et al., 2020, PNAS)

*7) L181: Do you use "precipitation scavenging" and "impact scavenging" with different meanings or the same meaning?*

**Response:** Thank you for your comment. They have different meanings. Precipitation scavenging refers to removal caused by precipitation, such as below-cloud washout, while impact scavenging refers to removal resulting from the collision between cloud water and aerosols within the cloud (Barrett et al., 2019; Choi et al., 2020). Scavenging includes in-cloud scavenging and precipitation scavenging, while in-cloud scavenging includes nucleation scavenging and impact scavenging. The primary difference between Bare BC and Coated BC lies in in-cloud scavenging, whereas differences in the precipitation scavenging process are negligible. Therefore, we modified the in-cloud scavenging algorithm. Apologies for any confusion caused. We have re-drawn Fig.1 and replaced Fig. 2 with a clearer Table 2, without mentioning precipitation scavenging.

[Figure]

● Internally mixed BC aerosol    ● Externally mixed BC aerosol    ● Scattering aerosols    ● Cloud drops

**Figure 1: The BC mixing state in the WRF-CMAQ model and the WRF-CMAQ-BCG model.**

**Table 2. Comparison of BC aerosol in major processes (Aitken and accumulation modes).**

| Processes \ Species | | BC (Original) | Bare BC (New) | Coated BC (New) |
|---|---|---|---|---|
| | **Emission** | Yes | Yes | No |
| | **BC Aging** | No Aging process | Bare BC is aged to Coated BC | |
| **Wet Deposition** | Impact scavenging | Yes | Yes | Yes |
| | Nucleation scavenging | Yes | No | Yes |
| | **Aerosol Optics** | Core-shell sphere | Homogeneous sphere | Core-shell sphere |

*8) L194: in direct radiative forcing -> in estimating direct radiative effect*

**Response:** Thanks for your suggestion. We have changed "in direct radiative forcing" to "in estimating direct radiative effect" in the revised manuscript.

*9) L221: Are all BC emissions treated as externally mixed particles?*

**Response:** Thank you for your comment. Yes, we consider freshly emitted BC to be externally mixed, consistent with the approach adopted by most atmospheric models. In our model, this external mixing assumption is used to represent the initial stage of BC aging, where BC particles are treated as separate entities. Over time, these particles may undergo processes like condensation, coagulation, and other interactions that lead to their final internal mixing state with other components. Even though we recognize that BC can form chain-like structures through aggregation, the interaction between BC and other aerosol

species remains relatively limited, especially when they are freshly emitted. Therefore, we kept this simplification in our model, as it sufficiently represents the typical behavior of BC emissions without bringing in unnecessary complexity or large uncertainties.

*10) L267-269: Figure 6a: Why is this spatial distribution obtained?*

**Response:** Thank you for your comment. This is related to the distribution of OH concentration. The areas with higher values reflect higher atmospheric oxidizing capacity and more active photochemical reactions, leading to a higher aging rate. We have added relevant explanations in the revised manuscript.

"The aging rate ($k$) and the aging timescale ($\tau$) are important variables to quantify the aging process. The standard WRF-CMAQ model lacks the capability to generate BC aging-related variables. In contrast, in the WRF-CMAQ-BCG model, spatiotemporal variations of the aging-associated variables in the BC aging process are related to the concentration of OH radicals. The areas with higher values reflect higher atmospheric oxidizing capacity and more active photochemical reactions, leading to a higher aging rate. Fig. 5(a) shows the aging rate…"

*11) L278-279: Again, are all BC emissions treated as externally mixed particles?*

**Response:** Thank you. Yes, please refer to our response to Comment 9 for the details.

*12) Figure 8: The unit of the vertical axis is unclear. This is only a qualitative evaluation, and a quantitative evaluation is needed.*

**Response:** Thanks for your suggestion. We have changed the unit of the vertical axis to "Normalized $NF_{\text{coated}}$". Due to the variability of BC mass concentrations and mixing states across different regions and times, a direct quantitative comparison is not feasible. Therefore, we normalized $NF_{\text{coated}}$ to facilitate quantitative comparison and updated the relevant text in the revised manuscript.

"Due to the variability of BC mass concentrations and mixing states across different regions and times, we normalized $NF_{\text{coated}}$ to facilitate quantitative comparison, as shown in Eq. (5). By comparing the simulated and observed Normalized $NF_{\text{coated}}$, it is evident that the BC mixing state reflects a temporal variation characteristic, with the proportion of Coated BC significantly increasing during the daytime.

$$\text{Normalized } NF_{\text{coated}} = \frac{NF_{\text{coated}} - NF_{\text{coated\_min}}}{NF_{\text{coated\_max}} - NF_{\text{coated\_min}}} \ , \tag{5}$$

Where $NF_{\text{coated\_min}}$ is the minimum value of $NF_{\text{coated}}$, $NF_{\text{coated\_max}}$ is the maximum value of $NF_{\text{coated}}$."

[Figure]

**Figure 8:** Daily average variation in simulated results: (a) Bare BC mass concentration ($M_{bare}$), (b) Coated BC mass concentration ($M_{coated}$), (c) Number fraction of Coated BC ($NF_{coated}$) and **(d) Normalized $NF_{coated}$** compared with other observational studies.

*13) L319-320: This description is probably incorrect. Aerosols are transported over long distances in a few days, so I think the speed of aging is not related to Fig. 9a. I think this is because in-cloud scavenging is not considered (only below-cloud scavenging is considered).*

**Response:** Thanks for your suggestion. We have revised the text to: "This distinctive distribution is attributed to the spatial distribution of Bare BC concentration, and the fact that Bare BC cannot be removed by nucleation scavenging."

*14) Figure 13: Why don't you show the time series plot for MAC?*

**Response:** We have added the time series plot for MAC in Figure 12. The original Figure 12 shows the time series of the absorption coefficient ($b_{abs}$) and BC mass concentration ($M_{BC}$). Since the MAC value is directly derived from the ratio of these two variables ($b_{abs}/M_{BC}$), we included it in Figure 12 and updated the relevant text in the revised manuscript.

[revised manuscript text omitted]

---

## Author Comment (AC3)

**Response to the comments of Editor (egusphere-2024-2372)**

*I advise the authors to provide an extended discussion considering all referees comments, especially considering one of the referee's criticism of novelty associated to BC ageing being previously modeled with similar methods.*

**Response:** We sincerely appreciate the editor's thoughtful reminder. We have addressed all the reviewers' comments, including the critique related to the novelty of modeling BC aging with similar methods. Ambiguity in our original manuscript might have led to misunderstandings of the reviewer. Through collaboration with the CMAQ development team at the Office of Research and Development (ORD), US EPA, this is the first time the BC aging process has been considered and tracked within the framework of CMAQ model and WRF-CMAQ two-way coupled model. Even though CMAQ model is a modal model, our approach of incorporating BC aging process in the model is different from other existing modal models, such as CAM5-MAM3 and CAM5-MAM7 models (Liu et al., 2012; 2016). Unlike these models, which introduce an additional mode to represent externally mixed BC, our approach takes a different strategy for representing BC mixing state and aging dynamics.

The CMAQ model, as a well-established air quality model, includes over a hundred aerosol species and hundreds of chemical reactions, representing a robust set of physical-chemical processes. Within the WRF-CMAQ model framework, our focus, we introduced two new species (Bare BC and Coated BC) to represent different BC aging states and we believe this is a more suitable method to incorporate the effects of BC aging. Secondly, we represented the aging process through a virtual chemical reaction and modified key processes influenced by different BC aging states, including wet deposition and aerosol optics. The aerosol optics algorithm, which is highly sensitive to particle size, was updated with recalculating size information accordingly. This method minimizes computational complexity while incorporating BC aging into the CMAQ model and WRF-CMAQ coupled model. Without compromising the accuracy of commonly simulated variables in the original model, this method introduces the capability to simulate BC mixing states, this method introduces the ability to simulate BC mixing states, the wet deposition of internal and external BC, and improves the accuracy of BC mass concentration and aerosol optics. Therefore, our approach offers a novel perspective on incorporating BC aging within a modeling framework. We have revised the manuscript to articulate clearly the novelty of our method (details can be found in the response AC2).

**References**

Liu, X., Easter, R. C., Ghan, S. J., Zaveri, R., Rasch, P., Shi, X., Lamarque, J.-F., Gettelman, A., Morrison, H., Vitt, F., Conley, A., Park, S., Neale, R., Hannay, C., Ekman, A. M. L., Hess, P., Mahowald, N., Collins, W., Iacono, M. J., Bretherton, C. S., Flanner, M. G., and Mitchell, D.: Toward a minimal representation of aerosols in climate models: description and

evaluation in the Community Atmosphere Model CAM5, Geosci. Model Dev., 5, 709–739, doi:10.5194/gmd-5-709-2012, 2012.

Liu, X., Ma, P. L., Wang, H., Tilmes, S., Singh, B., Easter R. C., Ghan, S. J., and Rasch, P. J.; Description and evaluation of a new four-mode version of the Modal Aerosol Module (MAM4) within version 5.3 of the Community Atmosphere Model, Geosci. Model Dev., 9(2), 505-522, doi:10.5194/gmd-9-505-2016, 2016.

---

## Author Response (AR2)

**Response to the comments of Editor (egusphere-2024-2372)**

*The authors revised their manuscript substantially addressing most reviewers concerns. While the BC aging topic has been extensively studied in the past decade and some of the scientific knowledge and processes included in this study is not new (which are some of the remaining major concerns from reviewers), it is new to the WRF-CMAQ coupled model. It is important to advance the WRF-CMAQ modeling capability on this aspect given the wide use of this model and thus I agree to continue considering the manuscript for publication on ACP. Authors, please revise the manuscript for technical correction according to some of the remaining reviewers comments:*

**Response:** We sincerely thank the editor for recognizing and supporting our work. Please find our point-by-point responses listed below. The reviewers' comments are in *Italic* followed by our responses and revisions (in blue). The modifications in the manuscript are highlighted in red.

**The reviewers' comments:**

*# The only one minor thing that requires a further clarification is their response to my previous specific comment #9: The authors mentioned that they used WRFDA to assimilate NAM data. What variables did they assimilate? How frequently did they assimilate (e.g., every 6 hours)? Why did they do the DA using NAM instead of observations? Did the authors mean "meteorological nudging" instead of DA? Some clarification will be useful.*

**Response:** Thank you for pointing out the need for clarification regarding our data assimilation approach. FDDA algorithm is part of the EPA physics package incorporated into the WRF structure. This algorithm is based on NAM data with a 3-hour frequency. NAM data were chosen for their spatial and temporal continuity across the simulation domain, ensuring consistent and high-quality boundary and initial conditions to better constrain the model dynamics. The control of nudging as well as the strength of nudging is done in the namelist. In the WRF-CMAQ coupled model scenario, the strength is set to very low value (Gilliam et al., 2012) to let the dynamics do its work. Horizontal winds (U and V), temperature (T), and water vapor mixing ratio (Q) are the nudging variables.

"Additionally, we applied the Four-Dimensional Data Assimilation (FDDA) technique within the WRF model framework to improve the meteorological fields during the simulation. Specifically, we used analysis nudging to assimilate with a 3-hour frequency. The assimilated data were sourced from the North American Mesoscale Forecast System (NAM), provided by the National Weather Service's National Centers for Environmental Prediction (NCEP). The nudging variables included horizontal winds (U and V), temperature (T), and water vapor mixing ratio (Q). In the WRF-CMAQ coupled model scenario, the nudging strength was set to a very low value (Gilliam et al., 2012) to minimize interference with the model dynamics while constraining the meteorological fields."

*# The sentence '(coarse mode aerosols without BC component are not considered in this study)' has been added. Are particles without BC explicitly considered in the accumulation and Aitken modes? As I pointed out in my previous comment, considering particles without BC is important. If particles without BC are not considered, BC coating is overestimated, and this has a large impact on the lifetime and optical properties of BC.*

**Response:** Thank you for your suggestions. This work primarily investigates the impact of BC aging process on BC aerosol. Consequently, non-BC particles (particles without BC) in the Aitken and accumulation modes are not included in our analysis. We acknowledge that this omission may introduce some bias into our results. In our recent work, we have explored the impact of non-BC aerosols on aerosol optics in the CAM6-MAM4 model (Chen et al., 2023). As illustrated in Fig. R1, the Mass Absorption Cross-section (MAC) values of aged BC simulated by CAM6-ABC have undergone significant changes. These modifications, which include adjusting the BC core size and distinguishing non-BC particles, among other effects, have resulted in a substantial decrease in MAC values from approximately 13 m² g⁻¹ in the original model to about 8 m² g⁻¹ in the revised version. Extrapolating from this trend (see Fig. R2), we estimate that the MAC values simulated by our aging model could potentially decrease further, from around 9 m² g⁻¹ to approximately 8 m² g⁻¹. It is important to note that WRF-CMAQ is a two-way coupled meteorology - air quality model with complete internal mixing of aerosols assumption. Consequently, the presence of non-BC particles may significantly influence various atmospheric processes beyond those directly related to BC aging. In light of these considerations, we intend to conduct further investigations into the role of non-BC particles within the WRF-CMAQ framework in our future research. We sincerely appreciate your invaluable input, which has prompted us to incorporate a relevant discussion in the revised manuscript.

"In our WRF-CMAQ-BCG model, we calculated the optics of Bare BC and Coated BC separately in the Aitken and accumulation modes, without accounting for the presence of scattering aerosols independently. This omission may result in an overestimation of the coating thickness, potentially introducing some bias into the optical simulation results (Chen et al., 2023)."

[Figure]

**Figure R1:** Comparisons of BC MAC (MAC$_{BC}$) between the default model (CAM6-MAM4) and the new model (CAM6-ABC) in different BC size distributions (coating thickness (CT)). (cited from Chen et al., 2023)

[Figure]

**Figure R2:** Schematic of the trend in the change of MAC values after considering non-BC particles (Note: Unlike the MAC results presented in the paper, this figure does not include Bare BC and Aitken mode).

*#Coagulation is important, so I do not recommend using a constant aging value with no reasons/discussion.(Please see Riemer et al., 2019 and references therein).*

**Response:** Thank you for your suggestion. As we mentioned in our previous response, we agree that coagulation is an important process in BC aging near sources. However, within the context of mesoscale modeling, the effects of coagulation on aerosol dynamics are generally considered to be of minor importance compared to condensation.

Riemer et al. (2010) demonstrated that even under extreme conditions highly conducive to coagulation in a supersaturated environment (with supersaturation levels ranging from 0.1% to 1%) with high concentration of aerosols, the impact of coagulation on BC aging remained less significant than that of condensation. This was evidenced by the stark contrast in aging time-scales: in condensation-dominated daytime environments, aging occurred much more rapidly (11 to 0.068 h) compared to coagulation-dominated nighttime environments (54 to 6.4 h). These findings underscore that condensation plays a more substantial role in accelerating the BC aging process. Zaveri et al. (2010) used the particle-resolved PartMC-MOSAIC model to study aerosol mixing state evolution in an idealized urban plume. Their results (Fig. R3) showed slightly higher mass concentrations without coagulation, indicating that coagulation only has a minor effect. Oshima et al. (2009) and Doran et al. (2008) also support this conclusion.

[Figure]

**Figure R3:** Comparison of aerosol mass concentration with coagulation (solid) and without coagulation (dashed). Evolution of bulk aerosol species in the urban air parcel as it is advected downwind over a period of 2 days. (cited form Zaveri et al., 2010)

Given the above studies, we have treated the condensation-related terms in the aging rate as fast-aging term and coagulation as a slow-aging term, simplifying it to a constant, as done in studies by Liu et al. (2011), Huang et al. (2013), Oshima and Koike (2013), and others. In the future, we will explore using variables to represent the rates of coagulation and condensation processes to obtain a more accurate BC aging rate. We have added the requested reasons/discussion in the description of Eq. (3) and cited the relevant papers.

"This study employs a BC aging module that quantifies the BC aging rate using an equation dependent on the concentration of OH radicals, as shown in Eq. (3). Condensation is considered through the setting of a fast-aging term, while coagulation is considered through a slow-aging term. Although coagulation can occur rapidly near sources, it does not play a dominant role within the context of mesoscale modeling compared to condensation (Doran et al., 2008; Oshima et al. 2009; Riemer et al., 2010; Zaveri et al., 2010). …

$$k = \beta\,[OH] + \alpha\,, \tag{3}$$

where $k$ represents the aging rate, $[OH]$ represents OH radical concentration. $\beta$ and $\alpha$ are assumed to be constant, with values $4.6 \times 10^{-12}$ cm$^3$ molecule$^{-1}$ s$^{-1}$ and $5.8 \times 10^{-7}$ s$^{-1}$, $\beta$ is estimated by assuming an e-folding aging timescale of 2.5 days for condensation, and $\alpha$ is estimated by assuming a 20 days e-folding lifetime for coagulation (Liu et al., 2011; Huang et al., 2013; Oshima and Koike, 2013). Indeed, this constant assumption does not account for the dynamic variations of the coagulation and condensation processes, which could introduce some bias."

**References**

Gilliam, R. C., Godowitch, J. M., and Rao, S. T.: Improving the horizontal transport in the lower troposphere with four dimensional data assimilation, Atmos. Environ., 53, 186-201, doi:10.1016/j.atmosenv.2011.10.064, 2012.

Chen, G., Wang, J., Wang, Y., Wang, J., Jin, Y., Cheng, Y., Yin, Y., Liao H., Ding, A., Wang S., Hao J., and Liu, C.: An aerosol optical module with observation-constrained black carbon properties for global climate models, J Adv. Model Earth Sy., 15, e2022MS003501. doi:10.1029/2022MS003501, 2023.

Liu, J., Fan, S., Horowitz, L. W., and Levy, H.: Evaluation of factors controlling long-range transport of black carbon to the Arctic, J. Geophys. Res. Atmos., 116(D4), doi:10.1029/2010JD015145, 2011.

Huang, Y., Wu, S., Dubey, M. K., and French, N. H. F.: Impact of aging mechanism on model simulated carbonaceous aerosols, Atmos. Chem. Phys., 13(13), 6329-6343, doi:10.5194/acp-13-6329-2013, 2013.

Oshima, N., and Koike, M.: Development of a parameterization of black carbon aging for use in general circulation models, Geosci. Model Dev., 6(2), 263-282, doi:10.5194/gmd-6-263-2013, 2013.

Zaveri, R. A., Barnard, J. C., Easter, R. C., Riemer, N., and West, M.: Particle‐resolved simulation of aerosol size, composition, mixing state, and the associated optical and cloud condensation nuclei activation properties in an evolving urban plume, J. Geophys. Res. Atmos., 115, D17, doi:10.1029/2009JD013616, 2010.

Riemer, N., M. West, R. A. Zaveri, and R. C. Easter: Estimating black carbon aging time scales with a particle resolved aerosol model, J. Aerosol Sci., 41, 143–158, doi:10.1016/j.jaerosci.2009.08.009, 2010.

Oshima, N., M. Koike, Y. Zhang, and Y. Kondo: Aging of black carbon in outflow from anthropogenic sources using a mixing state resolved model: 2. Aerosol optical properties and cloud condensation nuclei activities, J. Geophys. Res., 114, D18202, doi:10.1029/2008JD011681, 2009.

Doran, J. C., J. D. Fast, J. C. Barnard, A. Laskin, Y. Desyaterik, and M. K. Gilles: Applications of Lagrangian dispersion modeling to the analysis of changes in the specific absorption of elemental carbon, Atmos. Chem. Phys., 8(5), 1377–1389, doi:10.5194/acp-8-1377-2008, 2008.